# Photo-reversible amyloid nanoNETs for regenerative antimicrobial therapies

Qize Xuan[1,2,8], Hui Li[1,8], Yuan Gao[1,8], Xinchi Qiao[1], Yifan Feng[1], Xinyu Yu[1], Jiazhe Cai[1], Tonghui Jin[2], Bin Liu[2], Mohammad Peydayesh [2], Jiaqi Su [2], Peter Fischer [2], Ping Wang[3], Chao Chen [1] ✉, Jiangtao Zhou [2,4,5,6] ✉ & Raffaele Mezzenga [2,7] ✉

Drug-resistant bacterial infections, exacerbated by antibiotic resistance and biofilm resilience, disrupt tissue repair through dysregulated inflammation and impaired regeneration. Neutrophil extracellular traps (NETs) play a crucial role in endogenous immunity by entrapping and eliminating pathogens, inspiring the development of synthetic biomaterials that replicate this function. However, current synthetic NETs face challenges in complexity, biocompatibility, structural integrity and effectiveness. Here, we present a NETs-mimicking hydrogel composed of reversible lysozyme amyloid flexible nanofibrils (FFs) enabling pathogen elimination and tissue regeneration. The FFs therein self-assemble from natural egg-white lysozyme endowing these nanoNETs with bioactivity against pathogens, and when duly labeled to respond to near-infrared irradiation, they disassemble into unfolded lysozyme monomers with antimicrobial activity. Notably, the hydrogel disassembly is followed by the controlled release of pre-dissolved $Mg^{2+}$ ions, reprogramming macrophages toward a pro-regenerative phenotype and mitigating inflammation. In both murine and porcine models, these biocompatible nanoNETs demonstrate excellent antibacterial performance, accelerating healing of wounds infected by methicillin-resistant *Staphylococcus aureus* (MRSA). Moreover, these nanoNETs boost in-vivo healing of MRSA-infected periprosthetic joints, preserving osteogenic and regenerative microenvironments. These results build on the reversible nature of flexible amyloids to introduce stimuli-responsive biocompatible nanoNETs with significant potential for antimicrobial and regenerative therapies in bacterial-resistant infections.

Microbial infections remain a global health crisis, causing approximately 7.7 million of deaths annually and imposing a significant economic burden on healthcare systems[1]. This situation is further exacerbated by the rise of antimicrobial resistance (AMR) due to the abuse of antibiotics, rendering many conventional therapies ineffective[2]. The World Health Organization (WHO) has identified AMR as one of the top ten global public health threats[3], with multidrug-resistant pathogens such as methicillin-resistant *Staphylococcus aureus* (MRSA) becoming increasingly prevalent. Beyond their pathogenic effects, bacterial infections severely prevent tissue regeneration by forming biofilms that impair cell migration and angiogenesis. This further prolongs inflammation, disrupts the balance between pro-inflammatory and pro-regenerative signaling, and introduces bacterial toxins that damage host cells[4–6]. These multifaceted challenges demand innovative strategies that not only guarantee pathogen eradication, but also promote the restoration of the tissue microenvironments.

In the human body, the innate immune system serves as the first line of defense against pathogens, with neutrophils at the forefront[7]. One key strategy of neutrophils is the release of web-like neutrophil extracellular traps (NETs), composed of DNA, histones, myeloperoxidase (MPO), neutrophil elastase (NE), and other granular proteins. NETs physically trap and neutralize pathogens, thereby preventing their dissemination[8]. However, dysregulated or excessive NETs release can lead to tissue damage and chronic inflammation. The emergence of synthetic NET biomaterials demonstrated their potential to replicate NETs antimicrobial activity while avoiding detrimental effects[9]. Recent advances have highlighted various synthetic materials to mimic the structure and function of NETs to harness their pathogen-trapping and antimicrobial properties[10,11]. These biomimetic NET systems including DNA-based hydrogels[12,13] and synthetic polymer networks[14,15] can effectively capture bacteria, but often lack intrinsic antimicrobial activity and rely on exogenous agents, such as antibiotics or nanoparticles, limiting their adaptability and effectiveness in complex biological environments[16]. Peptide-based systems, on the other hand, demonstrated the advantages of synthesis simplicity, favorable physicochemical properties, and structure-functionality tunability[17–20], however, they often require complex chemical modifications to achieve stimuli responsiveness, raising concerns about their biocompatibility and safety[21]. These challenges underscore the need for novel NETs-mimicking materials that combine dynamic responsiveness, multifunctionality, and biocompatibility.

Lysozyme, one of the first identified host defense peptides (HDPs), is a key component of the innate immune system and is widely present in mammals, particularly in humans[22]. Lysozyme compromises the structural integrity of bacteria by hydrolyzing peptidoglycan in their cell walls[23], whereas the antimicrobial proteins in NETs, including neutrophil elastase and myeloperoxidase, primarily disrupt microbial membranes or interfere with metabolic processes[24–26]. These features suggest lysozyme as an appealing solution for constructing NETs-biomimetic biomaterials with intrinsic antimicrobial activity. However, despite extensive efforts to develop lysozyme-based biomaterials, a significant loss of bioactivity is often observed during fabrication due to their hydrolysis and aggregation[27,28]. For instance, native lysozymes assemble into amyloid rigid fibrils (RFs) in acidic and heating conditions after undergoing unfolding, hydrolysis, and misfolding processes, resulting in the loss of their enzymatic function[29,30].

To address these issues, we turn our attention to native hen-egg white lysozyme, leveraging on its distinctive capacity of forming reversible, flexible amyloid fibrils (FFs)[31,32]. Distinct from conventional RFs (Fig. 1a), these FFs are formed through a rapid heat shock at neutral pH, inducing partial unfolding without hydrolysis and thus preserving the entirety of lysozyme functional domains. In contrast to RFs possessing a rigid amyloid core, FFs feature a short β-sheet in the amyloid core[31] that loosely nano-confines partially-unfolded lysozymes curly and flexible amyloids with significantly lower rigidity. Remarkably, these FFs demonstrate thermal reversibility, disassembling into unfolded, yet active lysozyme monomers at 50 °C, and re-assembling upon reheating[32].

In this work, using thermally-responsive FFs as building blocks, we introduce a biomimetic system inspired by the dynamic behavior of NETs, to develop a stimuli-controlled "smart" nanoNET (i.e. NET based on reversible amyloid nanofibrils) for pathogen entrapment and inactivation, which serves in vivo as scaffold for wound healing and tissue regeneration. This nanoNET is fabricated through co-assembly of lysozyme FFs and magnesium chloride, forming a hydrogel incorporating indocyanine green (ICG) to enable photothermal responsiveness, a dye already approved by the Food and Drug Administration (FDA) (Fig. 1b). In the absence of stimuli, the FFs form a mesh of inactive flexible amyloid fibrils. Upon near-infrared (NIR) irradiation, the photothermal effect of ICG triggers the disassembly of FFs into

monomeric lysozymes, reactivating their bacteriolytic function and simultaneous release of $Mg^{2+}$ ions, which promote tissue repair by regulating macrophage polarization. This dynamic behavior not only replicates the natural formation and dissolution of NETs but also provides a tunable system for targeted pathogen control via non-invasive stimuli. Unlike conventional antimicrobial agents that exacerbate tissue damage through non-selective cytotoxicity, our lysozyme-based nanoNETs provide a dual benefit: a biocompatible scaffold supporting cell migration and tissue regeneration, and the confined bacteriolytic activity that preserves the surrounding regenerative microenvironments. In vivo studies confirmed that the biomimetic nanoNETs feature the desirable antibacterial efficacy and therapeutic potential in promoting the MRSA-infected wound healing of murine and porcine models, as well as the effective control of bone-implant infection in mice (Fig. 1c). Together, our findings establish a proof-of-concept for NETs-inspired materials based on biocompatible HDPs with controllable enzymatic activity. It is noted that our biomimetic engineering strategy focuses on functional emulation rather than structural replication, aiming to create a minimalist, HDP-based functional mimic that achieves the sequential "capture-and-kill" dynamic of NETs. The nanofibrous network acts as a physical trap for bacteria "capture", analogous to the DNA/protein fibers of native NETs. The photothermally-triggered release of bioactive lysozyme provides the antimicrobial "kill" function, analogous to the role of MPO, NE, and other effectors in native NETs. This study opens promising avenues for the development of next-generation biomaterials serving in resistant infections, antimicrobial therapies and regenerative medicine.

## Results and discussion

### Photothermal reversibility characterization of lysozyme flexible fibrils (FFs) nanoNETs

Atomic force microscopy (AFM) images (Fig. 2a and Supplementary Fig. 1) show the morphologies of lysozyme FFs and RFs, in the absence and the presence of ICG. FFs exhibit a short and curly shape whereas RFs feature a long and rigid morphology, with diameters of 2.5 and 7.8 nm, respectively. Notably, ICG modification induced no significant morphological differences (Fig. 2b). However, the persistence length (Fig. 2c), representing fibril rigidity, showed a notable increase in both fibrils upon ICG incorporation: FFs double from 60 to 120 nm while RFs increase from 6 to 10 μm. These findings indicate the successful incorporation of ICG molecules into both FFs and RFs[33], enhancing fibril stiffness with minimal changes in fibril diameter. Secondary structural investigation via circular dichroism (CD) and Thioflavin T (ThT) fluorescence assay (Supplementary Fig. 2) showed that FFs are rich in both α-helical and β-sheet contents, whereas RFs were predominantly composed of β-sheet structures. These results confirm that FFs are composed of partially unfolded lysozyme with preserved functional domains necessary to enzymatic activity, whereas RFs contain a robust β-sheet-rich amyloid core[31].

Subsequently, hydrogels composed of either FFs or RFs were fabricated by introducing magnesium chloride (Fig. 2d), which screen electrostatic charges by reducing the Debye length[34]. These hydrogels show tunable mechanical strength, proportional to lysozyme and $MgCl_2$ concentrations (Supplementary Figs. 3-4). Both ICG-incorporated RFs (RFs-ICG) and FFs (FFs-ICG) exhibited strong absorbance in the NIR region (700−850 nm), distinct from free ICG (Supplementary Fig. 5), possibly due to ICG binding to the amyloid β-sheets. Consequently, FFs-ICG hydrogels demonstrated excellent photothermal conversion efficiency, dependent on both NIR intensity and ICG concentration (Fig. 2e-f and Supplementary Fig. 6). Additionally, FFs-ICG hydrogels demonstrated superior stability and cycling performance compared to pristine ICG molecule (Fig. 2g and Supplementary Fig. 7), which is attributed to the homogeneous dispersion of ICG within FF-hydrogel, preventing ICG aggregation and subsequent photobleaching upon NIR irradiation[35].

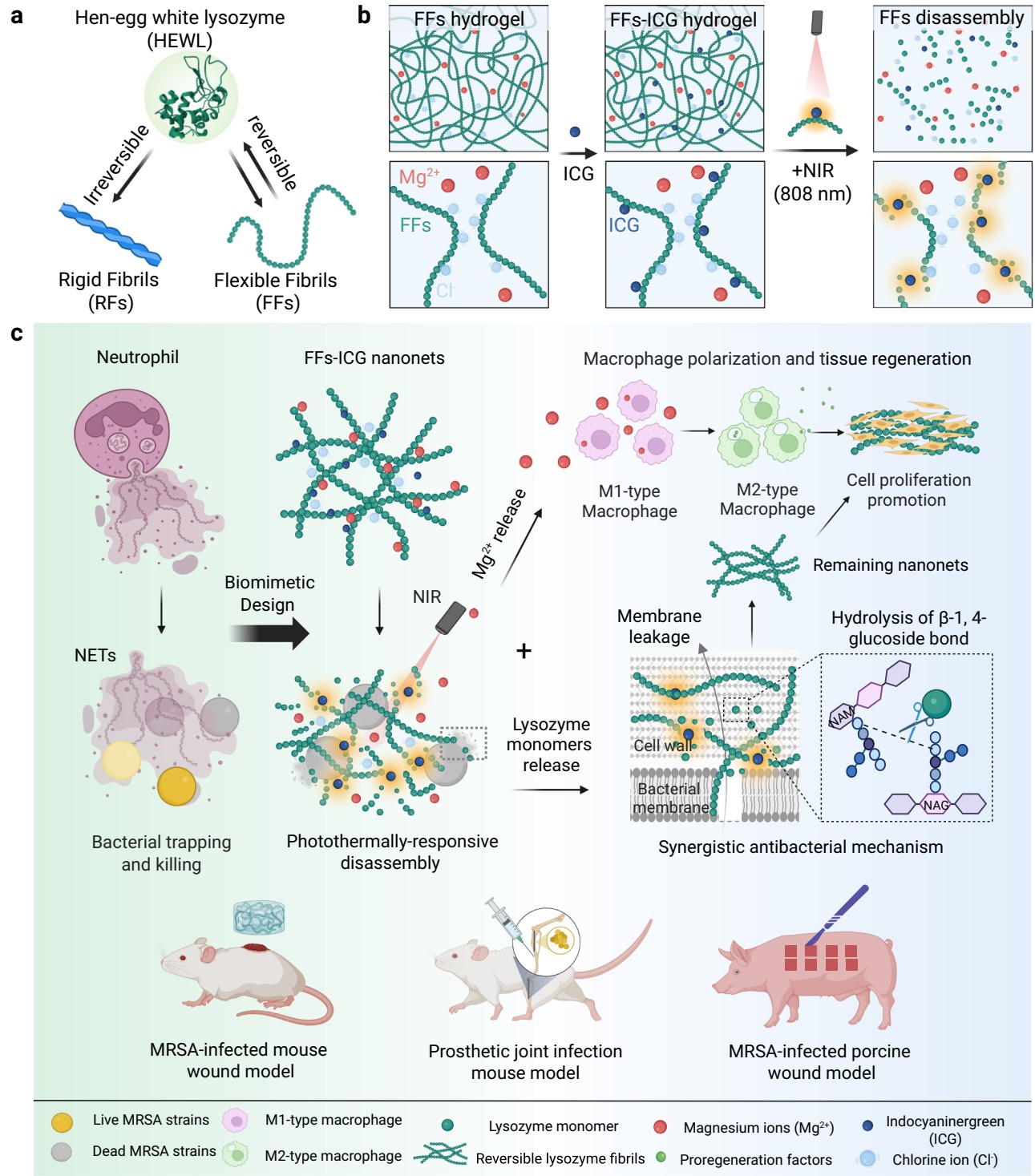

**Fig. 1 | Photothermally-reversible lysozyme amyloid fibril nanoNETs for adaptive antimicrobial therapy. a** Schematic illustrations of the fabrication of irreversible rigid fibrils and reversible flexible fibrils from hen-egg white lysozyme. **b** Mechanism of photothermal reversibility of FFs-ICG nanoNETs, releasing the native lysozymes and Mg²⁺. **c** Schematics of the mechanism of NETs-mimicking FFs-ICG nanoNETs for antibacterial and tissue regeneration. Upon NIR irradiation, the nanoNET releases bioactive lysozyme monomers that eliminate bacteria through membrane leakage and hydrolysis of cell wall components, and simultaneously releases Mg²⁺ promoting macrophage polarization and cell proliferation. The antibacterial efficacy and tissue regeneration of FFs-ICG nanoNETs are validated in three in vivo models, including MRSA-infected mouse and porcine skin wound, and the prosthetic joint infection mouse model. Created in BioRender. Xuan, Q. (2025) https://BioRender.com/v8dvned.

We further explored the morphological and structural transformations of FFs-ICG hydrogels induced by photothermal treatments. After 10 min of NIR irradiation, FFs-ICG hydrogels exhibit a distinct gel-solution transition (Supplementary Fig. 8) and FFs disassemble into smaller aggregates or monomers (Fig. 2h). Notably, these disassembled building blocks reassemble into FFs after 2 h, resulting in a solution-to-gel reverse transition. In contrast, both RFs and its hydrogels showed no significant changes during the heating-cooling process. These results highlight the thermal- and photothermal-responsive reversible behavior of FFs-based hydrogels. Moreover,

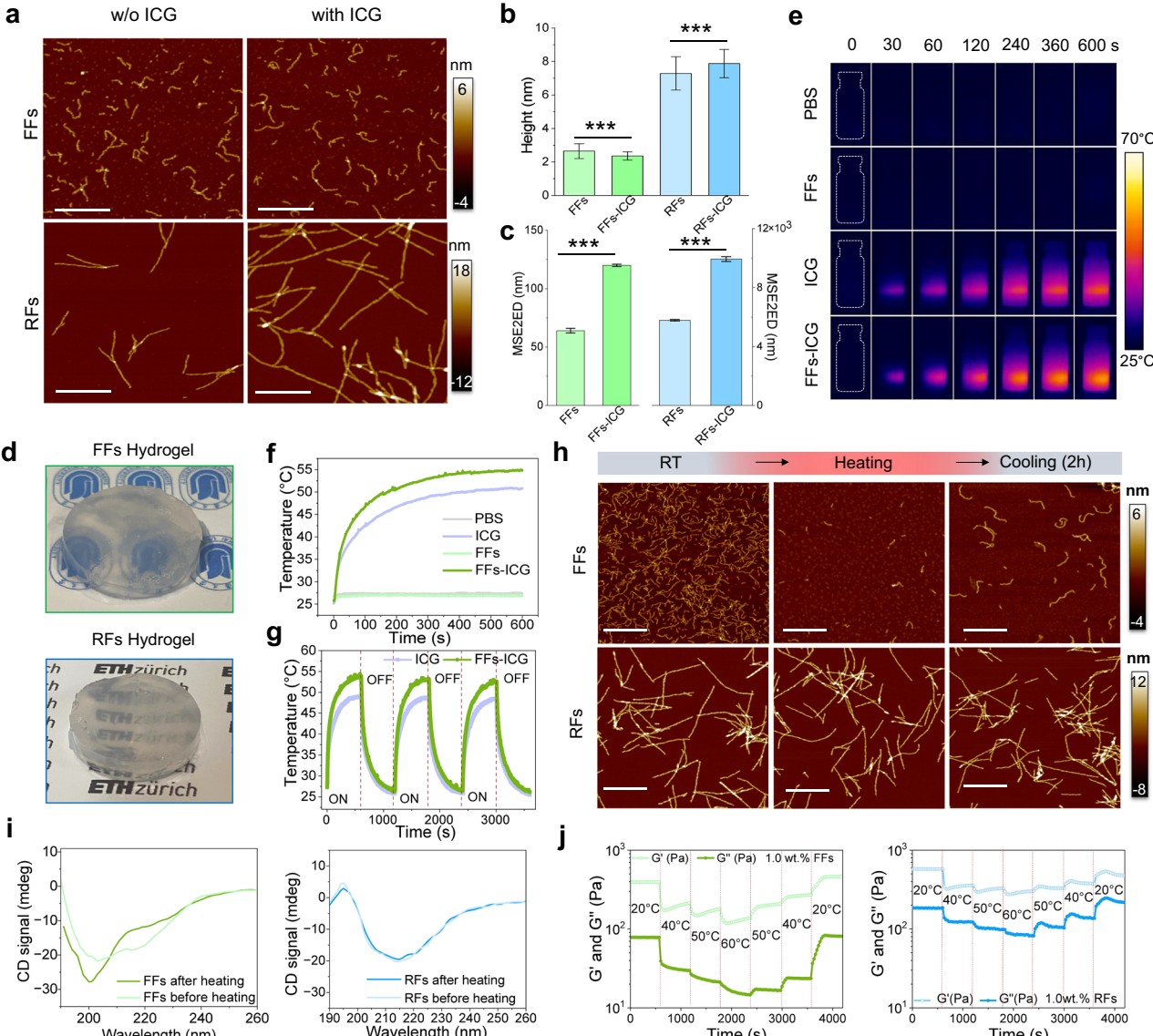

**Fig. 2 | Reversibility of flexible fibrils and FFs-ICG hydrogel. a** AFM images of lysozyme FFs and RFs in the absence and the presence of ICG (scale bar: 500 nm). Statistical analysis on the height (**b**) of FFs and RFs based on AFM investigation. A large number of fibrils ($n = 449$) were included in the height analysis. The data were expressed as mean ± standard deviation (S.D). Statistical significance was analyzed by one-way ANOVA using GraphPad Prism 8, followed by Tukey's post-hoc test for pairwise comparisons. Statistical significance was defined as \**p < 0.05*, \*\**p < 0.01*, and \*\*\**p < 0.001*. FFs *vs*. FFs-ICG ($p < 0.0001$), RFs *vs*. RFs-ICG ($p < 0.0001$). Fibril rigidity was studied by measuring the mean square end-to-end distance (MSE2ED) of their polymeric chain[47]. The morphological data of these fibrils were collectively fitted to obtain MSE2ED value (**c**) of FFs/FFs-ICG and RFs/RFs-ICG, with the error bar representing fitting error using the 2D worm-like chain model[48]. FFs vs. FFs-ICG ($p < 0.0001$), RFs *vs*. RFs-ICG ($p < 0.0001$). **d** Photo of FFs hydrogel and RFs hydrogel. Representative photothermal imaging (**e**) and the corresponding photothermal curve (**f**) of FFs, ICG, and FFs-ICG after 808 nm NIR irradiation. **g**, Comparison of ICG and FFs-ICG in the photothermal conversion stability. **h** AFM images of FFs and RFs (scale bar: 800 nm) under the heating and cooling treatments. **i** CD analysis of the secondary structural transition of FFs and RFs before and after heating at 60 °C. **j** Temperature-dependent rheological changes of FFs and RFs hydrogel (1.0 wt.% concentration) ranging from 20 to 60 °C. Source data are provided as a Source Data file.

CD spectroscopy (Fig. 2i) revealed a decrease in the negative peak at 218 nm and an increase at 198 nm of FFs upon heating, indicating a structural transition from β-sheet to random coil[31], whereas RFs exhibited no significant changes. Rheological study confirmed that RFs exhibited temperature-dependent decrease of mechanical strength, evidenced by the gradual decrease in both storage modulus (G') and loss modulus (G") with cyclic heating (Fig. 2j and Supplementary Fig. 9), due to gel melting[36]. On the other hand, FFs hydrogel showed a significantly more pronounced reduction of mechanical strength upon heating (Fig. 2j and Supplementary Fig. 10), suggesting a combined effect of gel softening and FF disassembly. This scenario is supported by the ThT-stained confocal imaging, as shown in Supplementary

Fig. 11, showing a significant decrease in fluorescence intensity of FFs after photothermal treatments, whereas RFs remained stable. Notably, both RFs and FFs hydrogel retain the gel state, maintaining G' > G" even at 60 °C, suggesting partial preservation of the architecture in their hydrogel networks.

### In vitro characterization for the bioactivity of flexible fibrils (FFs) nanoNETs
Lysozyme possesses intrinsic antibacterial activity. However, its applications are limited due to the tendency to aggregation and potential cytotoxicity at high concentrations[29,37]. We hypothesized that the formation of FFs could mitigate cytotoxicity, while preserving

intrinsic antibacterial activity. To confirm this, we first applied L-929 fibroblast in the assessment of FFs cytocompatibility (Fig. 3a). Results indicate that lysozyme monomers (1 wt.%) reduced cell viability to 50%, whereas both RFs and FFs maintain cell survival to 90%. This suggests that the linear alignment of lysozyme significantly mitigates the cytotoxic effect of native lysozyme. Live/dead-staining confocal imaging results verified the cytocompatibility of the FFs hydrogel (Supplementary Fig. 12). The concentration-dependent cytotoxicity in L-929 cells was also evaluated (Supplementary Fig. 13), demonstrating an excellent biocompatibility of the FFs hydrogel, even at high-concentration exceeding 1 wt.%. Similar results were observed in the cytotoxicity assays using human umbilical vein endothelial cells (HUVECs) (Supplementary Fig. 14). In addition, both FFs and RFs displayed a high level of hemocompatibility, maintaining the hemolysis ratio ranging from 0.1 to 1 wt.% concentration (Supplementary Fig. 15).

Unlike traditional protein nanofibrils that decrease their cytotoxicity by compromising biological activity[38], the FFs effectively align partially-unfolded active lysozyme, avoiding high cytotoxicity while retaining biological activity. Notably, this enables the controllable release of active lysozyme monomers upon photothermal-stimulation. This release was quantified by measuring lysozyme monomer concentration in the RFs and FFs solution before and after photothermal treatments (Supplementary Fig. 16). Consistent with the observation above (Fig. 2h), RFs released negligible amounts of monomers after 10 min at 60 ℃, whereas FFs exhibited a fivefold increase in lysozyme monomer concentration, compared to their release at 20 ℃ (Fig. 3b). The released monomers retained enzymatic bioactivity, linearly correlated with the initial FFs concentrations (Fig. 3c). Upon NIR irradiation, FFs-ICG exhibited a significant increase in enzymatic activity, whereas RFs, FFs, and RFs-ICG remained with low enzymatic functionalities (Fig. 3d). These results confirm that the nanoconfined bioactive lysozyme in the FFs can be released upon photothermal treatment (Fig. 3e), while irreversible RFs remain intact upon heat treatment due to their stable misfolded β-sheet-rich structure derived from the fully-unfolding state.

To demonstrate the pathogen-trapping and killing capacity of our nanoNETs, we utilized MRSA and *Pseudomonas aeruginosa* (*P. aeruginosa*, a highly motile Gram-negative bacteria known for its swimming and swarming motility) as model strains. As shown in Supplementary Fig. 17, our nanoNETs can achieve the effective physical entrapment of both MRSA and *P. aeruginosa* within the fibrous network, visually confirmed by scanning electron microscope (SEM) images, which offer direct morphological evidence of bacterial capture, regardless of bacterial motility. We have also experimentally confirmed that both strains (MRSA and *P. aeruginosa*) exhibit negative surface charges, as measured by zeta potential in Supplementary Fig. 18. This characteristic enables them to be effectively captured through electrostatic adsorption by the positively charged nanoNETs[29,38]. We found that the lysozyme monomers exhibited a high antimicrobial efficiency ($89.2 \pm 2.5\%$), outperforming RFs ($62.1 \pm 5.5\%$) and FFs ($71.9 \pm 0.7\%$) (Fig. 3f and Supplementary Fig. 19). RFs-ICG achieved a higher antibacterial rate ($81.5\% \pm 3.8\%$) partially increased by ICG, whereas FFs-ICG demonstrated the highest antibacterial performance ($95.4\% \pm 0.6\%$) within low-power NIR irradiation ($0.3\,W/cm^2$). Notably, this low photothermal power allows to accurately assess the differences in antibacterial activity between RFs and FFs. These results are supported by the live/dead bacterial staining images (Fig. 3g). Further, SEM imaging (Fig. 3h) shows the MRSA membrane disruption, and the leakage of N-acetyl-beta-D-glucosidase (NAG) and $K^+$ after treatments (Supplementary Fig. 20-21), indicating the increased bacterial membrane permeability as the antibacterial activity. Additionally, FFs-ICG nanoNETs exhibited an antimicrobial efficacy of $86.2 \pm 4.3\%$ for *P. aeruginosa*, and $89.8 \pm 4.7\%$ for *Candida albicans* (a typical fungal strain) (Supplementary Fig. 22), demonstrating the broad-spectrum antibacterial

efficacy of nanoNETs beyond Gram-positive bacteria. These results indicate our FFs possess a synergistic effect against bacteria: NIR-induced FFs partially disassemble and release active lysozyme for bacteria degradation, while the residual FFs-ICG nanoNETs maintain the function of pathogen entrapment through the absorption-contact mechanism as conventional RFs[39,40], collectively providing direct evidence to support the "trap-and-kill" mechanism of the nanoNETs.

In addition to lysozyme release, we also quantified the $Mg^{2+}$ ion release from the hydrogel. Following NIR irradiation, $Mg^{2+}$ concentrations in RFs and FFs solutions increased approximately six- and eight-fold, respectively, compared to pre-irradiation levels (Fig. 3i). The release of magnesium chloride indicates the partial dissociation of hydrogel network and the reduction of mechanical strength. On the other hand, magnesium ions are known to promote macrophage polarization from M1 to anti-inflammatory M2 phenotype and facilitate tissue regeneration[41]. To confirm this, we used Raw 264.7 cells as the macrophage model to assess the immunomodulatory capacity of our nanoNETs, with IL-4 as the positive control for M2 polarization. We observe a clear correlation between $Mg^{2+}$ release and M2-type polarization (Fig. 3j–l), with the highest M2 polarization ratio ($36.7 \pm 1.3\%$) in FFs-ICG group due to the photothermal-responsive characteristics. Dysregulated or excessive native NETs release can lead to tissue damage and chronic inflammation, whereas our nanoNETs can avoid these detrimental effects and harnessing the beneficial antimicrobial mechanism. The data above provide definitive evidence for the role of released $Mg^{2+}$ from nanoNETs in macrophage polarization to prevent the occurrence of these adverse outcomes. Additionally, we collected supernatant from macrophages treated with FFs-ICG, which was then added into the medium and co-incubated with L-929 cells. The result revealed that this supernatant significantly enhanced L-929 cells proliferation by 39.91% compared to the control group (Supplementary Fig. 23), suggesting the presence of reparative cytokines in the supernatant and thus highlighting the functional role of M2-polarized macrophages in promoting cellular growth and tissue regeneration. Collectively, FFs-ICG nanoNETs provides a multifunctional platform for anti-infection therapy: acting as the "host defense network" to trap bacteria, releasing bioactive lysozyme upon irradiation for antimicrobial effects, and enabling $Mg^{2+}$ release for promoting tissue repair via macrophage polarization regulation.

## In vivo evaluation of FFs-ICG nanoNETs in an MRSA-infected murine wound model

To evaluate the in vivo therapeutic efficiency of FFs-ICG hydrogel, we established a murine model of MRSA-infected wound healing (Fig. 4a). MRSA-infected wounds were treated with PBS, lysozyme, RFs, FFs, ICG, RFs-ICG, and FFs-ICG. Quantitative analysis revealed that NIR-irradiated FFs-ICG treatment achieved a nearly complete wound closure (approximately 100%) by day 14, significantly outperforming the PBS group ($71.5\% \pm 8.2\%$) (Fig. 4b, c and Supplementary Fig. 24a). Correspondingly, bacterial enumeration demonstrated a > 95% reduction in MRSA viability in the FFs-ICG group (Fig. 4d). By contrast, RFs-ICG exhibited both relatively lower healing and antibacterial efficacy than the FFs-ICG, while lysozyme monomers demonstrated strong antibacterial activity (Fig. 4d) but impaired wound healing (Fig. 4b and Supplementary Fig. 24b), likely due to their cytotoxic effects.

Histological analysis of Hematoxylin & eosin (H&E) stained sections (Fig. 4e and Supplementary Fig. 25a) further demonstrated accelerated regeneration in the FFs-ICG group compared to other groups, characterized by complete dermis and epidermis reconstruction and nascent hair follicle formation. Immunohistochemical detection of Ki-67 and Giemsa staining confirmed enhanced cellular proliferation and antimicrobial effect in the FFs-ICG treated tissues (Fig. 4f–g and Supplementary Fig. 25b–d). Notably, the substantial

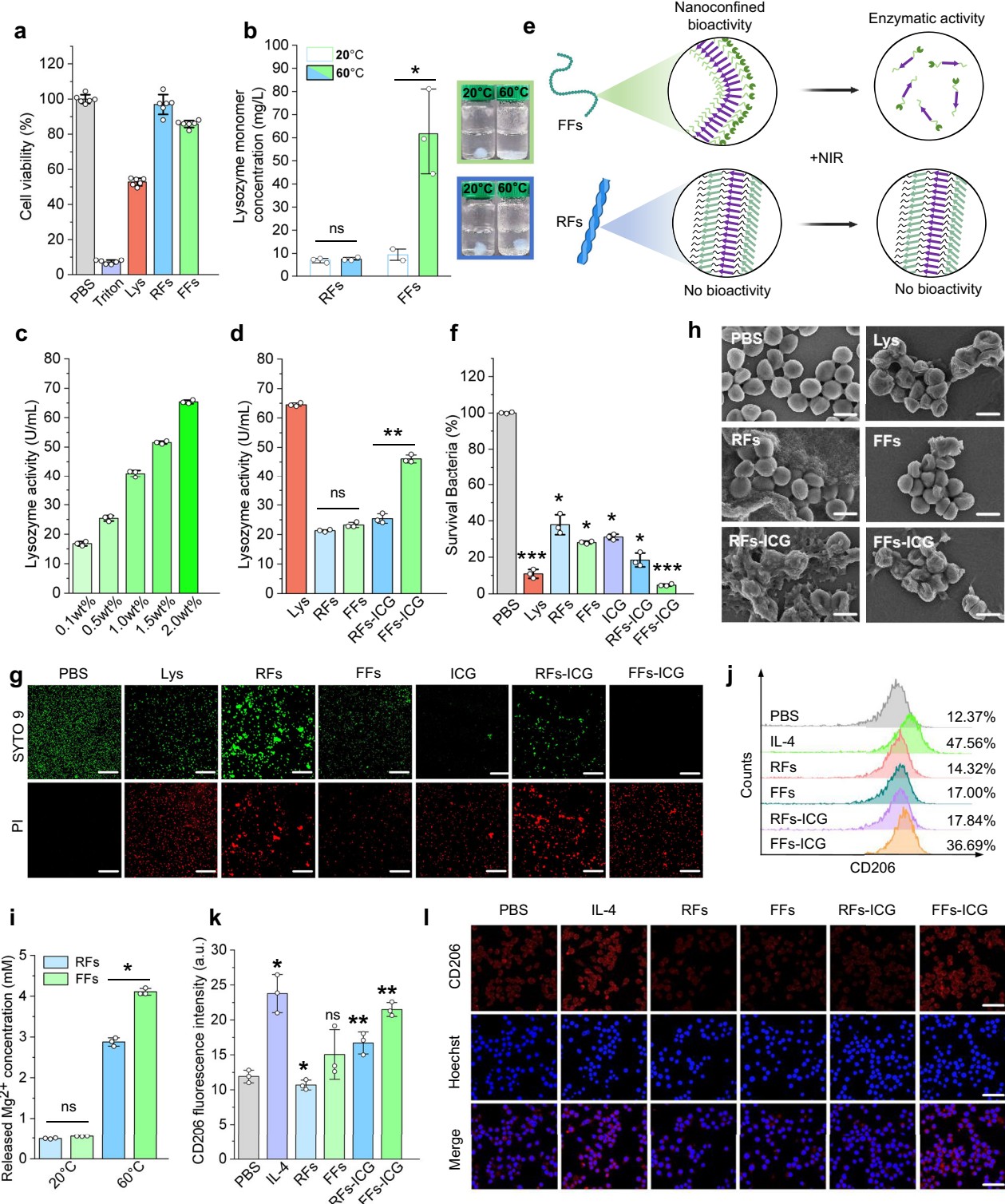

Mg$^{2+}$ release induced by FFs-ICG under NIR exposure significantly promoted the macrophage polarization to M2 phenotype, particularly in the dermal layer, which contributed to sustained tissue regeneration (Fig. 4h). In contrast, PBS-treated wounds exhibited only granulation tissues formation, while FFs and RFs groups showed partial dermal regeneration (Fig. 4e and Supplementary Fig. 25). These groups also demonstrated inferior performances in cell proliferation, antibacterial efficacy, and immunomodulation (Supplementary Fig. 25), consistent with in-vitro results. We further performed quantitative PCR validation on wound tissues and observed significant upregulation in the mRNA

expression of the genes related to key reparative cytokines and growth factors in the FFs-ICG group, including *Vegfa, Hif1α, Il10, and Pdgfb*, compared to the PBS control (Supplementary Fig. 26). This provides direct molecular evidence that FFs-ICG actively promotes a pro-regenerative transcriptional program within the wound micro-environment, perfectly aligning with the observed functional outcomes of accelerated healing. Furthermore, the therapeutic benefit of our FFs-ICG nanoNETs extends to the established infections (3 days) with mature biofilms (Supplementary Fig. 27). This significantly enhances the clinical relevance of our platform, as it demonstrates

**Fig. 3 | In vitro characterization of FFs-ICG nanoNETs in antibacterial activity and macrophage polarization. a** Cytocompatibility assays ($n = 6$) of lysozyme monomers, FFs, and RFs using methyl thiazolyl tetrazolium (MTT) assay. **b** Quantification ($n = 3$) of released monomers from lysozyme FFs and RFs at 20 °C and 60 °C. RFs 20 °C vs. RFs 60°C ($p = 0.3398$), FFs 20 °C vs. FFs 60°C ($p = 0.0196$). Lysozyme activity assays ($n = 3$) of FFs-ICG at different concentrations after NIR irradiation (**c**) and RFs/FFs before and after NIR irradiation (**d**). RFs vs. FFs ($p = 0.0583$), RFs-ICG vs. FFs-ICG ($p = 0.0031$). **e** Schematic illustration for RFs/FFs bioactivity and thermally-induced disassembly. **f** Antibacterial efficiency assays ($n = 3$) of Lys, RFs, FFs, RFs-ICG and FFs-ICG against MRSA using CFU counting methods. Lys ($p = 0.0001$), RFs ($p = 0.0013$), FFs ($p < 0.0001$), ICG ($p < 0.0001$), RFs-ICG ($p = 0.0004$), FFs-ICG ($p < 0.0001$) vs. PBS control. SEM images (scale bar: 1 μm) (**g**) and live/dead bacteria staining confocal images (scale bar: 100 μm) (**h**) of MRSA

strains after different treatments. **i** Inductively-coupled plasma mass spectrometry (ICP-MS) assays ($n = 3$) of released $Mg^{2+}$ concentration before and after heating treatments. RFs 20 °C vs. FFs 20 °C ($p = 0.0056$), RFs 60 °C vs. FFs 60 °C ($p = 0.0017$). Flow cytometry analysis ($n = 3$) (**j, k**) and confocal microscopic images (scale bar: 50 μm) (**l**) of Raw 264.7 cells cultured with RFs, FFs, RFs-ICG, and FFs-ICG, with PBS and IL-4 as the negative and positive control, respectively. IL-4 ($p = 0.0149$), RFs ($p = 0.0475$), FFs ($p = 0.1702$), RFs-ICG ($p = 0.0036$), FFs-ICG ($p = 0.0063$) vs. PBS control. The data were expressed as mean ± standard deviation (S.D). Statistical significance was analyzed by one-way ANOVA using GraphPad Prism 8, followed by Tukey's post hoc test for pairwise comparisons. Statistical significance was defined as $*p < 0.05$, $**p < 0.01$, and $***p < 0.001$. Source data are provided as a Source Data file.

efficacy not just as a prophylactic or early intervention, but as a potential treatment for ongoing, resistant infections. Collectively, these results establish that FFs-ICG effectively eradicates MRSA infections while simultaneously promoting cell proliferative and tissue regenerative processes.

A comprehensive biosafety evaluation, including histological assessment of major organs and blood biochemical analysis, confirmed the excellent in vivo biocompatibility of all investigated biomaterials (Supplementary Fig. 28). Also, the hematological analysis demonstrated that the treatments of these prepared biomaterials, including our FFs-ICG nanoNETs, did not elicit significant systemic inflammation or disrupt leukocyte homeostasis, supporting their favorable safety profiles (Supplementary Fig. 29). More importantly, our FFs-ICG nanoNETs also exhibited a low risk of inducing thrombosis (including platelet activation and thrombus formation), supporting their safety for proposed topical applications (Supplementary Fig. 30). The lack of thrombogenicity is likely due to the biocompatible nature of lysozyme and the flexible, non-rigid structure of the FFs, which is less prone to mechanically triggering coagulation cascades compared to stiff fibrous materials. To further verify photothermal-triggered FFs disassembly in vivo, we performed the analysis of ThT-stained wound tissue sections from FFs-ICG and RFs-ICG groups on day 1 (Fig. 4i–j). Both groups exhibited high fluorescence signal without NIR irradiation. However, following irradiation, FFs-ICG-treated tissues showed significantly reduced ThT fluorescence, suggesting in vivo FFs disassembly. Partial post-irradiation reassembly was evidenced from the slight fluorescence recovery. In contrast, RFs-ICG maintained constant strong fluorescence, consistent with in vitro observations. These observations underline the dual functional mechanism of FFs-ICG hydrogel: (i) Positively charged FFs initially mimic NETs to entrap MRSA while NIR-triggered monomer release enables bacterial eradication; and (ii) Subsequent FFs reassembly templates an ordered scaffold that promotes cell adhesion and proliferation[37]. Additionally, our results facilitate to distinguish the contribution of photothermal effects and the release of bioactive lysozyme/$Mg^{2+}$. The photothermal effect (from ICG) provides a strong, non-specific antibacterial action, while the FFs provides a passive "trap" function and a biocompatible scaffold. The synergistic effect was achieved in FFs-ICG + NIR, and the photothermal effect triggers the on-demand release of the specific biological "kill" agent (lysozyme) and the pro-healing agent ($Mg^{2+}$), while the fibrous network provides a protective scaffold that mitigates the potential collateral damage of pure PTT and supports cell growth. It is the photothermal (ICG)-controlled dynamic functionality of the nanoNETs, the ability to convert from a passive trap to an active antimicrobial and regenerative factory, that delivers the optimal therapeutic outcome. This hypothesis was further verified by the results from a sterile (non-infected) murine full-thickness wound healing model (Supplementary Fig. 31), which confirmed that FFs-ICG nanoNETs platform possessed intrinsic pro-regenerative properties that operate independently of its antimicrobial function.

## In vivo evaluation of FFs-ICG nanoNETs in an MRSA-infected periprosthetic joint infection murine model

Based on the potent antibacterial and tissue-regenerative capabilities of the FFs-ICG hydrogel, we further investigated its therapeutic potential against endogenous infections using a periprosthetic joint infection model in mice (Fig. 5a). Mice were administered a total dose of 500 mg/kg nanoNETs formulations on day 1. As a distinct feature of the infected wound model, the formation of a biofilm emerges as the predominant survival strategy for MRSA in periprosthetic joint infection (PJI) mice model, rather than the existence of free-floating bacterial cells within the bone tissues[42]. Therefore, the implants with biofilm were extracted and analyzed using SEM, and bacterial burden on the implant and the surrounding bone tissues was quantified by CFU counting on day 7 to evaluate antibacterial efficacy. SEM images (Fig. 5b and Supplementary Fig. 32a) revealed near-complete biofilm disruption on the implants in the FFs-ICG group, with only deformed bacteria remnants visible. Correspondingly, CFU analysis confirmed that FFs-ICG achieved a significant bacterial clearance efficacy of $99.7 ± 0.2\%$ for peri-implant bacteria and $94.6 ± 1.1\%$ antimicrobial efficiency for MRSA-infected tibias (Fig. 5c, d). In addition, the diameter of the infected knee greatly decreased in PJI mice after FFs-ICG treatment (Fig. 5e), indicating reduced knee swelling and alleviation of infection-induced inflammation. These findings demonstrate the effective and broad-spectrum antibacterial activity of our FFs-ICG nanoNETs, independent of the survival mode (free-floating or biofilm) and the pathogen source (endogenous or exogenous).

Orthopedic implant infections often lead to bone resorption and periprosthetic osteolysis[43,44]. To investigate whether FFs-ICG nanoNETs could prevent infection-induced peri-implant osteolysis, we performed micro-CT analysis of tibias osteolysis (Fig. 5f and Supplementary Fig. 32b). Compared to the non-infected group, a severe osteolysis was observed around the implant with increased intertrabecular spacing and reduced bone mass in the PBS group, whereas FFs-ICG group exhibited preserved bone mass and maintained normal bone architecture. Quantitative analysis further (Fig. 5g, h) revealed significantly higher trabecular thickness (TB.TH) and peri-implant bone volume fraction (bone volume/total volume, BV/TV) in the ICG, RFs-ICG, and FFs-ICG groups, compared to PBS group. The highest efficacy was observed in FFs-ICG treated group, with values approaching those in non-infected mice. Interestingly, MRSA-infection also reduced joint bending angle (Fig. 5i), indicating impaired mobility, whereas FFs-ICG treatment effectively restored the joint flexibility to the level of non-infected mice, outperforming other treatment groups. Taken together, these results indicate that the FFs-ICG nanoNET not only is effective in implant-associated infections but it also prevents purulent arthritis and protects bone integrity.

Orthopedic implant-associated infections often require a treatment approach that extends beyond bacterial eradication to alleviate pain and preserve joint function. To assess their impact on mobility[44], we investigated the effects of FFs-ICG on the walking ability of mice with implant-associated infections through gait analysis (Fig. 5a). Mice

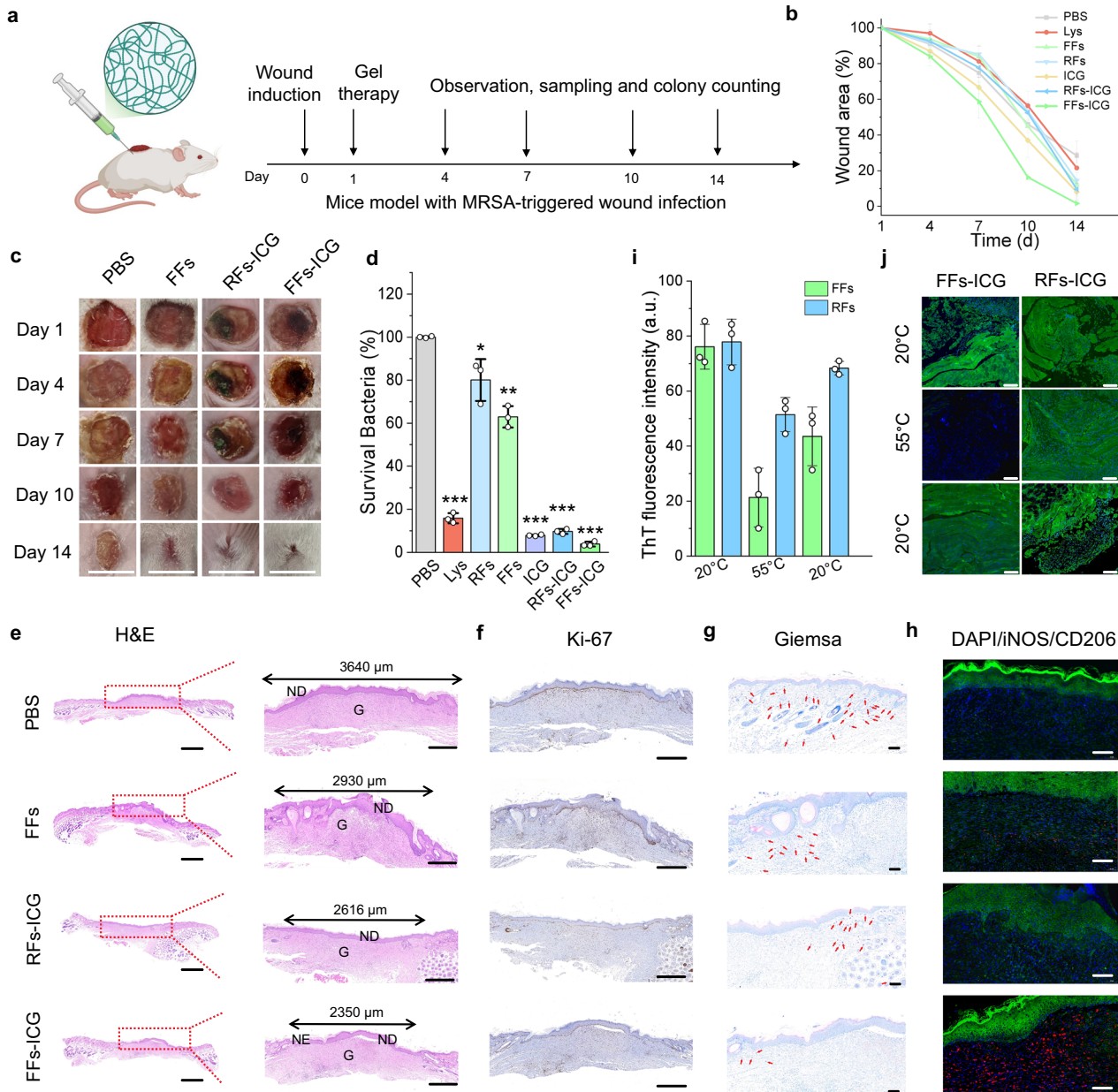

**Fig. 4 | In vivo therapeutic efficacy evaluation of FFs-ICG hydrogels in an MRSA-infected murine wound model. a** Timeline of the treatment and evaluation process of FFs-ICG hydrogel in an MRSA-infected wound healing mice model. Created in BioRender. Xuan, Q. (2025) https://BioRender.com/e9mjxve. Wound area quantification (*n* = 5) (**b**) and the corresponding representative macroscopic wound images (Scale bar: 1 cm) (**c**) of MRSA-infected mice after various treatments. **d** Quantification of MRSA survival rate (*n* = 3; data are presented as individual points) in the infected skin tissues after distinct treatments using CFU counting method. Lys (*p* = 0.0001), RFs (*p* = 0.0331), FFs (*p* = 0.0025), ICG (*p* < 0.0001), RFs-ICG (*p* < 0.0001), FFs-ICG (*p* < 0.0001) *vs*. PBS control. Representative H&E staining (**e**), Ki-67-staining (**f**), and Giemsa-staining (**g**), iNOS and CD206-staining (**h**) images of infected tissues from various treatment groups on day 14 for the analysis of

dermis and epidermis regeneration, cell proliferation, survival bacteria, and macrophage polarization, respectively. In Fig. 4e, g represents granulation tissue, and NE represents newly formed epidermis, and ND represents newly formed dermis. In Fig. 4g, the red arrows represent the residual bacteria. the fluorescence intensity quantification (**i**) (*n* = 3; data are presented as individual points) and the corresponding representative ThT-staining images (**j**) of wounds for the analysis of in vivo thermal-induced disassembling behaviors of FFs-ICG hydrogel. Scale bars: (**e**) 2 mm (left), 500 μm (right); (**f–h**) 500 μm; (**h**) 100 μm; (**j**) 200 μm. The data were expressed as mean ± standard deviation (S.D). Statistical significance was analyzed by one-way ANOVA using GraphPad Prism 8, followed by Tukey's post-hoc test for pairwise comparisons. Statistical significance was defined as *\*p* < 0.05, *\*\*p* < 0.01, and *\*\*\*p* < 0.001. Source data are provided as a Source Data file.

---

in the PBS and RFs-ICG groups displayed irregular and scattered footprints, indicating their impaired mobility (Fig. 5j and Supplementary Fig. 32c). In contrast, mice in the FFs-ICG group demonstrated well-defined footprints similar to those in the non-infected group, indicating their restored walking ability. Three-dimensional footprint reconstruction results (Fig. 5k and Supplementary Fig. 32d) further confirmed a normalized force distribution between soles and toes in

the FFs-ICG and non-infected group, while other treated groups (Lys, RFs, FFs, ICG, and RFs-ICG) showed incomplete foot engagement. Furthermore, significant improvements in support time, average intensity, and average speed (Fig. 5i–o) were observed in the FFs-ICG group, compared to the PBS group. These results highlight the efficacy of FFs-ICG in mitigating periprosthetic osteolysis and restoring joint functional mobility.

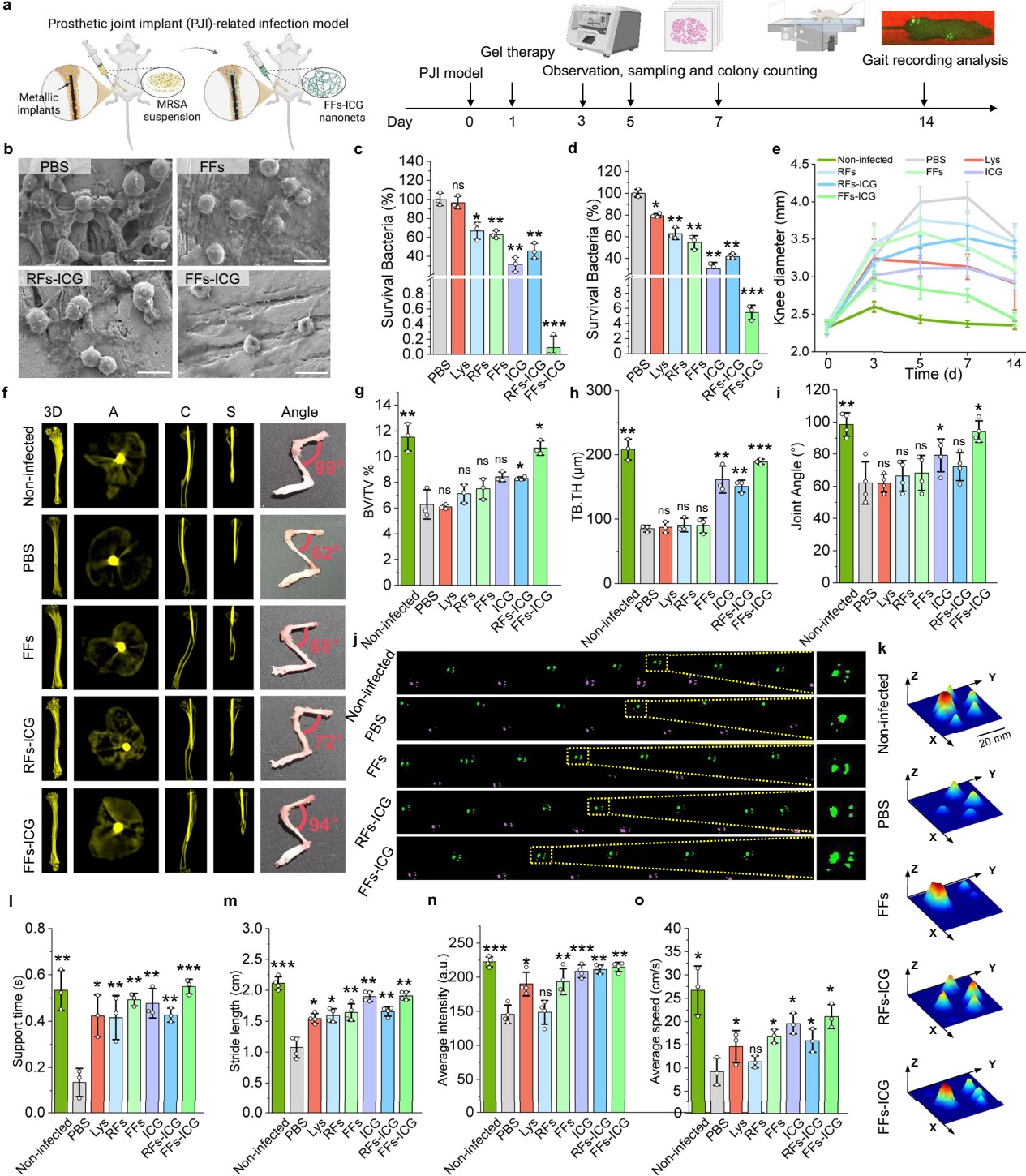

## In vivo evaluation of FFs-ICG nanoNETs in an MRSA-infected porcine wound healing model

To further examine the translational potential of FFs-ICG nanoNETs, we employed a porcine wound healing model, which closely recapitulates human skin architecture and re-epithelialization dynamics during tissue repair. Porcine skin shares key structural and functional similarities with human skin, including analogous dermal and epidermis layers, subcutaneous adipose tissue, sparse hair follicles distribution, and the presence of rete ridges[45,46]. As shown in Fig. 6a, eighteen square dermal wounds (1.5 cm × 1.5 cm × 0.6 cm) were bilaterally created on the porcine dorsum. Following a 4 h MRSA inoculation period (except for the non-infected group), wounds

were randomly assigned to various treatment groups. To minimize locational healing bias, the assignment was arranged in a randomized and crosswise manner. After treatments, the wounds were protected with sterile cotton gauze and a custom-made fenestrated jacket to prevent exogenous bacterial contamination. By day 15, the FFs-ICG-treated group demonstrated significantly enhanced wound closure efficacy compared to other treated groups (Fig. 6b, c and Supplementary Fig. 33a), and its residual wound areas were even smaller than those in non-infected controls. This superior regenerative and healing performance is understood to result from the cell proliferation-promoting properties of our FFs-ICG nanoNETs. Furthermore, the quantitative analysis revealed the lowest bacterial

**Fig. 5 | In vivo therapeutic efficacy evaluation of FFs-ICG hydrogel in an MRSA-infected prosthetic joint implants (PJI) model in mice. a** Schematic timeline to assess the therapeutic efficiency of FFs-ICG hydrogel in the MRSA-infected PJI model. Created in BioRender. Xuan, Q. (2025) https://BioRender.com/e9mjxve. SEM images of MRSA biofilm (**b**) on the surface of knee prosthesis at day 14. (Scale bars: 1 μm) CFU quantification ($n = 3$) of the joint peri-implant tissues (**c**) and bone (**d**) at day 14 in the PJI model. Lys ($p = 0.2910$), RFs ($p = 0.0102$), FFs ($p = 0.0022$), ICG ($p = 0.0057$), RFs-ICG ($p = 0.0047$), FFs-ICG ($p = 0.0007$) *vs.* PBS control in (**c**). Lys ($p = 0.0109$), RFs ($p = 0.0055$), FFs ($p = 0.0024$), ICG ($p = 0.0028$), RFs-ICG ($p = 0.0015$), FFs-ICG ($p = 0.0004$) *vs.* PBS control in (**d**). **e** Diameter of infected knee joints in PJI mice at days 0, 3, 5, 7, and 14 ($n = 3$ for each time point). **f** Axial, coronal, sagittal, and 3D reconstruction micro-CT images 4 weeks after MRSA infection. The mouse tibias are shown in dark yellow, and the implant is shown in light yellow. Quantitative micro-CT analysis including the bone volume/total volume (BV/TV) (**g**), and trabecular thickness (TB.TH) (**h**) ($n = 3$; data are presented as individual points). Non-infected ($p = 0.0017$), Lys ($p = 0.4128$), RFs ($p = 0.2015$), FFs ($p = 0.1380$), ICG ($p = 0.0546$), RFs-ICG ($p = 0.0483$), FFs-ICG ($p = 0.0210$) *vs.* PBS control in (**g**). Non-infected ($p = 0.0047$), Lys ($p = 0.3736$), RFs ($p = 0.1981$), FFs ($p = 0.3106$), ICG ($p = 0.0090$), RFs-ICG ($p = 0.0062$), FFs-ICG ($p < 0.0001$) vs. PBS control in (**h**). **i** Quantitative analysis of mouse knee-bending angle 4 weeks after infection (n = 4; data are presented as individual points). Non-infected ($p = 0.0049$),

Lys ($p = 0.4772$), RFs ($p = 0.1855$), FFs ($p = 0.2831$), ICG ($p = 0.0083$), RFs-ICG ($p = 0.1757$), FFs-ICG ($p = 0.0207$) *vs.* PBS control. **j–k** Gait analysis showing actual mice walking trajectory and footprint imaging. Green for left hind footprints; pink for right hind footprints (**j**). 3D reconstruction of left hind paw of mice in each group, with color-coded pressure intensity of the mouse's footprints (**k**). Quantitative gait analysis and the parameters: support time (**l**), stride length (**m**), average intensity (**n**) (average pressure on the runway), and average speed (**o**) ($n = 4$, data are presented as individual points). Non-infected ($p = 0.0081$), Lys ($p = 0.00394$), RFs ($p = 0.0026$), FFs ($p = 0.0016$), ICG ($p = 0.0039$), RFs-ICG ($p = 0.0068$), FFs-ICG ($p = 0.0008$) *vs.* PBS control in (**l**). Non-infected ($p = 0.0009$), Lys ($p = 0.0125$), RFs ($p = 0.0165$), FFs ($p = 0.0086$), ICG ($p = 0.0017$), RFs-ICG ($p = 0.0029$), FFs-ICG ($p = 0.0014$) *vs.* PBS control in (**m**). Non-infected ($p = 0.0005$), Lys ($p = 0.0183$), RFs ($p = 0.3766$), FFs ($p = 0.0069$), ICG ($p = 0.0010$), RFs-ICG ($p = 0.0018$), FFs-ICG ($p = 0.0033$) *vs.* PBS control in **n**. Non-infected ($p = 0.0104$), Lys ($p = 0.0329$), RFs ($p = 0.1638$), FFs ($p = 0.0182$), ICG ($p = 0.0368$), RFs-ICG ($p = 0.0319$), FFs-ICG ($p = 0.0323$) *vs.* PBS control in (**o**). The data were expressed as mean ± standard deviation (S.D.). Statistical significance was analyzed by one-way ANOVA using GraphPad Prism 8, followed by Tukey's post hoc test for pairwise comparisons. Statistical significance was defined as *$p < 0.05$, **$p < 0.01$, and ***$p < 0.001$. Source data are provided as a Source Data file.

survival rate in the FFs-ICG group (Fig. 6d, e), confirming their robust in vivo antibacterial activity.

Histological evaluation using H&E and Masson's staining demonstrated enhanced tissue regeneration in the FFs-ICG group, showing complete restoration of dermis and epidermis architectures, and well-organized collagen deposition, in addition to the smallest wound area (Fig. 6f, g and Supplementary Fig. 33b, c). Giemsa and Ki-67 immunohistochemistry analysis further validated the dual functionality of FFs-ICG nanoNETs, highlighting both robust antibacterial action and enhanced cell proliferative activity (Fig. 6h–i and Supplementary Fig. 33d–f). Immunofluorescence analysis of CD206 (Fig. 6j–k and Supplementary Fig. 33g) indicated a significantly elevated ratio of M2-phenotype macrophage in the FFs-ICG group compared to controls, indicating its potent immunomodulatory and anti-inflammatory effects. Additionally, comprehensive biosafety assessment, including the histological analysis of major organs and the blood biochemistry analysis, confirmed the excellent in vivo biocompatibility of these lysozyme-based nanofibrils (Supplementary Fig. 33h, i). Together, these results demonstrate that FFs-ICG nanoNETs promote wound healing through a multifunctional mechanism combining: (i) NETs-like antibacterial mechanism, (ii) anti-inflammatory modulation via macrophage polarization, and (iii) stimulation of cellular proliferation.

This work introduces lysozyme reversible amyloid fibrils as an original and powerful class of stimuli-responsible scaffolds for therapeutic tissue regeneration in wounds affected by bacterial infections. The escalating prevalence of antibiotic resistance demands innovative antimicrobial strategies that surpass conventional therapeutic paradigms. The developed photothermally switchable lysozyme-based nanoNETs system recapitulates natural NETs immune mechanisms enabling effective infection control and tissue regeneration upon non-invasive triggers, such as NIR. The resulting outstanding performance in tissue regeneration of these nanoNETs occurs via the dual release of bioactive lysozyme monomers, which facilitate site-specific bactericidal activity, and $Mg^{2+}$ ions, which modulate macrophage polarization, thus mitigating immune dysregulation. The potential of this approach was validated on three preclinical animal models, considering both soft tissue and orthopedic implant infections, demonstrating a significant reduction in bacterial infection, accelerated epithelialization, biofilm prevention, and joint function restoration. By leveraging natural immune strategies combined with photothermal activation, the biomimetic approach proposed here bridges host defense mechanisms with adaptable material design, a concept which may be further extended to

other bioactive proteinaceous compounds -such as defensins and myeloperoxidase- greatly expanding the scope of next-generation antimicrobial therapies and regenerative medicine.

## Methods

### Ethics statement

All animal experiments were conducted in accordance with Chinese legislation on the Use and Care of Research Animals (Document No. 55, 2001), and institutional guidelines for the Care and Use of Laboratory Animals established by the Shanghai University Animal Studies Committee, and this committee approved the experiments (ECSHU 2025-014).

### Lysozyme amyloid fibrils fabrication

Lysozyme FFs and RFs were obtained following our previous reports[31,32]. Briefly, the commercial hen egg white lysozyme (HEWL, Sigma Aldrich L6876) was dissolved in Milli-Q water at 20 mg/mL at room temperature for 3 h. Freshly prepared 1,4-Dithiothreitol (Sigma, D0632) at the concentration of 1 M and NaCl solutions (1 M) were added to reach a final protein solution, whose pH was then adjusted to pH 7 prior to incubation. The FFs were prepared by incubating the solution for 5 min at 90 °C with a magnetic stirring at 300 rpm. The RFs were prepared by incubating the solution for 3 h at 90 °C with a magnetic stirring at 300 rpm. Immediately after incubation, the sample was cooled in an ice bath for 30 min and stored at 4 °C prior to use.

### RFs/FFs-based hydrogel preparation and $Mg^{2+}$/monomers release measurements

The lysozyme amyloid fibrils hydrogel was prepared by mixing 0.5 mL of 2 wt.% RFs/FFs with 0.5 mL of $MgCl_2$(1 M) and ICG (1 mg/mL). The hydrogel was transferred into a sample vial, and 1 mL of deionized water was added. The RFs/FFs hydrogel was heated in a 60 °C water bath for 5 min, after which 1 mL of the supernatant was collected for analysis, with a non-heated group serving as the control. The released $Mg^{2+}$ concentration was measured using inductively coupled plasma mass spectrometry (ICP-MS, Agilent 8900, USA). The released content of lysozyme monomers was quantitatively analyzed using a standard curve acquired by ultraviolet/visible spectrophotometry (UV/Vis, UV-5100 Japan Hitachi).

### Lysozyme activity assays of released monomers

The as-prepared hydrogel was transferred into a sample vial, and 1 mL of deionized water was added. The RFs/FFs hydrogel was heated in a 60 °C water bath for 5 min, and 1 mL of the supernatant was collected,

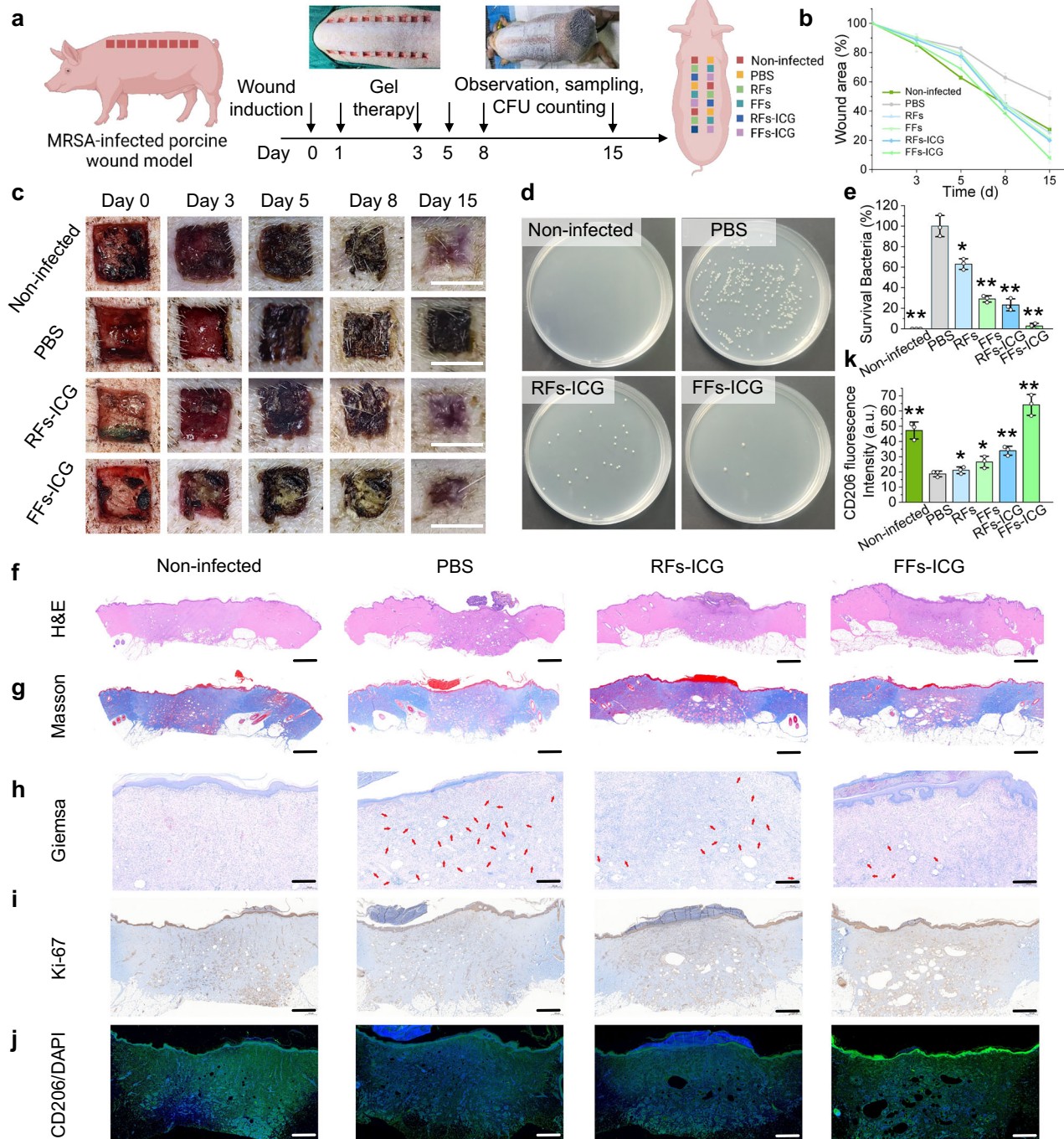

**Fig. 6 | In vivo therapeutic efficacy evaluation of FFs-ICG hydrogel in an MRSA-infected porcine wound model. a** Schematic timeline outlining the experimental design for assessing the therapeutic efficiency of FFs-ICG hydrogel in an MRSA-infected porcine wound model. Created in BioRender. Xuan, Q. (2025) https://BioRender.com/e9mjxve. Wound area quantification ($n = 3$) (**b**) and the corresponding representative wound images (Scale bar: 1.5 cm) (**c**) of MRSA-infected Bama pig after various treatments. The CFU plate images (**d**) and statistical analysis of MRSA survival rates (**e**) ($n = 3$; data are presented as individual points) in the infected skin tissues after different treatments. Non-infected ($p = 0.0018$), RFs ($p = 0.0270$), FFs ($p = 0.0054$), RFs-ICG ($p = 0.0070$), FFs-ICG ($p = 0.0023$) vs. PBS control. Representative H&E staining (scale bar: 1 mm) (**f**), Masson-staining (scale bar: 1 mm) (**g**), Giemsa-staining (Scale bar: 1 mm, top; 200 μm, down) (**h**), and Ki-67-staining (scale bar: 500 μm) (**i**) images of porcine wounds in different treatment

groups on day 15 for the analysis of dermis and epidermis regeneration, collagen deposition, survival bacteria, and cell proliferation, respectively. In Fig. 6h, the red arrows represent the residual bacteria. Representative CD206-staining images (scale bar: 500 μm) (**j**) and the corresponding quantification (**k**) ($n = 3$; data are presented as individual points) of porcine wounds for the analysis of macrophage polarization in different treatment groups on day 15. Non-infected ($p = 0.0084$), RFs ($p = 0.0225$), FFs ($p = 0.0472$), RFs-ICG ($p = 0.0010$), FFs-ICG ($p = 0.0022$) vs. PBS control. The data were expressed as mean ± standard deviation (S.D). Statistical significance was analyzed by one-way ANOVA using GraphPad Prism 8, followed by Tukey's post hoc test for pairwise comparisons. Statistical significance was defined as $*p < 0.05$, $**p < 0.01$, and $***p < 0.001$. Source data are provided as a Source Data file.

with the non-heated group, pure lysozyme monomer (Lys), and phosphate-buffered saline (PBS) as controls. Subsequently, 50 μL of the prepared test solution were subjected to perform lysozyme activity measurement according to the manufacturer's protocol of the EnzChek Lysozyme Assay Kit (E-22013).

### In vivo healing efficacy evaluation of in an MRSA-infected wound model of mice

Six-week-old healthy female BALB/c mice were randomized into seven groups of at least five mice per group, and they were shaved at the back. Afterwards, 1% sodium pentobarbital solution was used for the anesthesia of mice, and 1 cm-diameter wounds were created on the back of these mice using a skin perforator. Immediately, 50 μL of MRSA (ATCC 43300) suspensions ($1 \times 10^6$ CFU/mL) were injected and inoculated with wounds for 24 h. Then, these mice received different treatments according to their groups (PBS, Lys, RFs, FFs, ICG, RFs-ICG, and FFs-ICG). The mice in the ICG, RFs-ICG, and FFs-ICG groups were treated using near-infrared ray (NIR) irradiation (808 nm, 0.3 W/cm$^2$) photothermal therapy for 5 min post-injection. Wound status was monitored in the next two weeks, and photographs were taken daily to record the wound area. On day 7, the wound area was disinfected using 75% alcohol. Bacteria from the wounds were dipped using sterile cotton balls and added to the liquid medium and cultured to the logarithmic phase of the control bacteria (OD of 0.6–0.8), after which the samples were spread in agar plates using the gradient dilution method. The experiments were replicated three times for each group. On day 14, the skin tissues at the infected wound sites were dissected from the mice to perform the histological analysis. Hematoxylin and eosin (H&E) staining, Giemsa staining, Ki-67, and iNOS/CD206 immunofluorescence staining analysis of the dissected skin tissues were used to assess the tissue regeneration, survival bacteria, cell proliferation, and macrophage polarization in the infected wounds of mice. The major organs and blood of these mice were collected for the in vivo biosafety evaluation. For thioflavin T (ThT) staining analysis of assembly behaviors of fibrils before and after photothermal treatments, the back tissue of treated mice was taken, and stained with 40 μM ThT solution for 20 min, rinsed twice with PBS after the end, and observed at 488 nm using a CLSM (Zeiss, FV3000).

### In vivo therapeutic efficacy evaluation in a periprosthetic joint infection (PJI) model of mice

The MRSA (ATCC 43300) was cultured overnight at 37 °C and subsequently adjusted to an optical density (OD) of 0.8 at 600 nm using PBS prior to use. The PJI model of mice was fabricated according to a previous study[44]. Six-week-old healthy female BALB/c mice were randomly assigned to eight experimental groups, including Untreated, PBS, Lys, RFs, FFs, ICG, RFs-ICG, and FFs-ICG. These mice were anesthetized using 1% sodium pentobarbital, and their knees were disinfected with 75% ethanol. A longitudinal incision was made to expose the tibia, and a 29 G insulin needle (med-e-plus, China) was inserted into the tibia as the bone implants. The surgical sites were sutured, and a bacterial suspension (50 μL, $1 \times 10^6$ CFU/mL) was injected into the knee joints (except in the non-infected group). The next day, these mice (except the non-infected group) were injected with a total dose of 500 mg/kg of therapeutic agents (Lys, RFs, FFs, ICG, RFs-ICG, and FFs-ICG) or an equal volume of PBS on day 1. The mice in the ICG, RFs-ICG, and FFs-ICG groups were treated using near-infrared ray (NIR) irradiation (808 nm, 0.3 W/cm$^2$) photothermal therapy post-injection. On day 14 post-infection, the joint bone tissues and implants were collected, and bacterial burden was assessed by colony-forming unit (CFU) counting. The implants were placed in a 6-well plate and fixed with 2.5% glutaraldehyde at 4 °C overnight. Next, the implants were placed in different concentrations of ethanol solutions for dehydration (50%, 60%, 70%, 80%, 90% and anhydrous ethanol) for 10 min, freeze-dried, coated with platinum and then observed the biofilm of the

implant surfaces using a bio-scanning electron microscope (ZEISS Sigma 300, Germany). The tibial bone samples were harvested for micro-computed tomography (micro-CT) analysis (Skyscan 1172, Bruker Micro-CT, Germany) at a resolution of 9 μm when four weeks post-infections. The cancellous bone volume to total volume ratio (BV/TV) and trabecular thickness (TB.TH) were evaluated using the CTAn software (Skyscan, Bruker Micro-CT, Germany). Gait analysis experiments were performed on PJI model mice one week after infections. The mice were placed on the runway of the gait analyzer and ran from start to finish. High-speed camera at the bottom of the device was used to shoot rat footprints in 120 fps, 1/4 CCD. Support time, stride length, average intensity (average pressure on the runway) and average speed were analyzed by gait analysis software. According to the pressure of each foot contacting with the ground, 3D reconstruction of footprints was performed using gait analysis software.

### In vivo evaluation of infected wound healing efficacy in a porcine model

A female Bama pig (~25 kg weight) was selected for this experiment, which was performed in SJTU Biotechnology Experimental Co. Ltd, and its Animal Studies Committee approved this experiment (JDLL-P-20240907). Upon achieving full anesthesia, the pig was positioned laterally, and the dorsal region of pig was cleansed with 75% ethanol. The entire body of the pig was covered with a sterile surgical drape, exposing only the back. Eighteen full-thickness dermal wounds with identical size (1.5 cm long × 1.5 cm wide × 0.5 cm deep) were created on the dorsum, spacing 1.5 cm apart in a grid-like arrangement, located between the scapula and iliac crest on both side of the porcine back. Each square wound was marked using a sterile surgical marker. The epidermis was carefully exercised using skin forceps and double-edged surgical scissors. This process was repeated until all wounds were created successfully. Photographs of each wound on day 1 were taken, and the exact dimensions were measured and recorded using a calibrated surgical ruler. These wounds were then inoculated with 50 μL of MRSA (ATCC 43300) suspensions ($1 \times 10^6$ CFU/mL) for another 4 h. Subsequently, these wounds were assigned to one of six experimental groups, including Untreated, PBS, RFs, FFs, RFs-ICG, and FFs-ICG group, with three wounds ($n = 3$) for each group. To reduce the influence of the location of the wound on its healing efficiency, the wounds in different groups were distributed crosswise among each other. For these groups containing ICG (RFs-ICG, and FFs-ICG), NIR-induced photothermal treatment was applied using 808 nm (0.5 W/cm$^2$) light for 10 min. After these treatments, the wounds were covered with sterile cotton yarn and a custom-made fenestrated jacket was fitted into the pig to prevent the infection by exogenous bacteria. Wound exudates were collected on day 1, day 5, and day 8 post treatments, and the live bacterial counts were determined through the method of colony-forming unit (CFU) counting. Wound healing status was monitored and recorded on days 1, 5, 8, 15, and 21 post treatments. During the final treatment period, the pig was euthanized to collect wound tissues and perform histological analysis, including H&E and Giemsa staining. Macrophage polarization was assessed through immunohistochemical staining slice analysis at the infected sites. In these slice analyses, the normal skin of the Bama pig was set as the control. Additionally, the major organs and blood samples were harvested for biosafety evaluation.

### Statistics and reproducibility

All experiments and micrographs (AFM images, SEM images, CLSM pictures, and tissue section slicing) were performed in biological triplicate at minimum unless otherwise stated. The data were expressed as mean ± standard deviation (S.D). Statistical significance was analyzed by one-way ANOVA using GraphPad Prism 8, followed by Tukey's post hoc test for pairwise comparisons. Statistical significance was defined as *$p < 0.05$, **$p < 0.01$, and ***$p < 0.001$, and the notes of

special labels and detailed sample size are all given in the related experiments.

## Reporting summary

Further information on research design is available in the Nature Portfolio Reporting Summary linked to this article.

## Data availability

All data that support the findings of this study are available within the article and the Supplementary Information. Source data are provided with this paper Source data are provided with this paper.

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

## Acknowledgements

This work was sponsored by the National Natural Science Foundation of China (No. 42125706 for H.L., 42522711 for C.C., 42477424 for C.C., and 42407582 for Q.X.), the Natural Science Foundation of Shanghai (22ZR1415400 for C.C.), the Shanghai Rising-Star Program (23QC1400500 for C.C. and 24YF2712200 for Q.X.), the Shanghai Pujiang Program (24PJD038 for Q.X.), the Postdoctoral Fellowship Program of CPSF (GZB20250330 for Q.X.), and the China Postdoctoral Science Foundation (2024M761917 for Q.X.). Q.X. acknowledges financial supports from the China Scholarship Council (202206740021). The authors thank the guidance of Junming Lu and Dr. Yujie Hua from Shanghai Ninth People's Hospital Affiliated to Shanghai Jiao Tong University School of Medicine in the establishment and evaluation of porcine model. The authors thank Dr. Feng Jiang and Prof. Hao Shen from Shanghai Sixth People's Hospital Affiliated to Shanghai Jiao Tong University School of Medicine for the guidance of bacterial culture and testing. The authors also thank the Biorender website (www.biorender.com) for the preparation of schematics in this work.

## Author contributions

J.Z., Q.X. and R.M. conceptualized and designed the study. Q.X., Y.G., C.C. and J.Z. performed the experiments. X.Q., Y.F., X.Y. and J.C. participated in the in vivo experiments. Q.X., J.Z., Y.G., T.J., B.L., M.P., J.S., P.F., C.C., P.W., H.L. and R.M analysed the results. Q.X., J.Z., Y.G. and R.M. wrote the manuscript. H.L., C.C., P.W. and R.M supervised the project. All authors participated in the revision of manuscript.

## Funding

## Competing interests

The authors declare no competing interests.

## Additional information

¹Institute for Environmental Pollution and Health, School of Environmental and Chemical Engineering, Shanghai University, Shanghai, PR China. ²Department of Health Sciences and Technology, ETH Zürich, Schmelzbergstrasse 9, Zürich, Switzerland. ³Department of Bioproducts and Biosystems Engineering, University of Minnesota, St Paul, MN, USA. ⁴Department of Food Science and Technology, National University of Singapore, 2 Science Drive 2, Singapore, Singapore. ⁵Bezos Centre for Sustainable Protein at the National University of Singapore, 2 Science Drive 2, Singapore, Singapore. ⁶National University of Singapore (Suzhou) Research Institute, 377 Linquan Street, Suzhou Industrial Park, Jiangsu, China. ⁷Department of Materials, ETH Zurich, Wolfgang-Pauli-Strasse 10, 8049, Zürich, Switzerland. ⁸These authors contributed equally: Qize Xuan, Hui Li, Yuan Gao. ✉e-mail: chaochen@shu.edu.cn; jtzhou@nus.edu.sg; raffaele.mezzenga@hest.ethz.ch

