## [Transparent Peer review file · Nature Communications]

Photo-reversible Amyloid NanoNETs for Regenerative Antimicrobial Therapies

Corresponding Author: Professor Raffaele Mezzenga

Version 0:

Reviewer comments:

Reviewer #1

(Remarks to the Author)

The authors present an innovative reversible and stimuli-controlled hydrogel with antimicrobial activity and acting as scaffold for wound healing, with an in-depth in-vitro analysis as well as proof-of-concepts in in-vivo animal models. The study presents impressive results on the use of functional protein assembly to design network materials to fight current issues of bacterial resistance to antibiotics.

The work is technically sound, and the conclusions are well supported by the presented data. Enough detail is given for the reproducibility of the study. The reviewer suggests publication in Nature Communications, with only the following minor points to be addressed:

- 1) Figure 1 is generally clear, but it would benefit from some extra labelling of the panels to describe the molecular mechanisms of action of the hydrogel.
- 2) In Figure 2b, the authors show that ICG does not significantly affect fibril morphology, it would be helpful to indicate the number of fibrils analysed and use a statistical test to support further this conclusion.
- 3) In Fig. 2c it is not clear the meaning of the ordinate. The authors should comment also on the origin of the error in the persistence analysis.
- 4) The authors use in the text the sentence "while in irreversible RFs lysozyme is buried into irreversible aggregates.". While intuitively correct, the reviewer would suggest clarifying that the lysozyme is unfolded to form the fibrillar structure.
- 5) Could the authors describe more in detail which is the molecular origin of the disassembly of the FFs fibrils upon thermal treatment?
- 6) Which NIR power was used in the in-vivo experiments? How much in depth could go for further extension of this powerful approach?

Reviewer #2

(Remarks to the Author)

Review Photo-reversible Amyloid NanoNETs

The article presents a highly innovative approach for applying anti-microbial therapies by means of using flexible amyloid fibers that formed reversibly. Then innovative power lies in the combination of choosing the right protein that is forming amyloid fibers reversibly. In other words: the authors have found a route to mitigate cytotoxicity, while preserving intrinsic antibacterial activity of the proteins added. Even more specialistic, the flexible fibers play two roles, one of entrapment of bacterial species that do harm, and the degradation by the proteins forming reversibly these flexible fibers. I think this is a truly amazing and very unexpected finding. The authors combine dynamic responsiveness, multifunctionality and

biocompatibility.

Amazingly, although a heat shock treatment is taking place to induce fiber formation, the fiber formation is reversible, and the proteins remain enzymatically active. The reversibility allows for photo-induction of disintegration and subsequent efficacy.

The impact of the work is large as it is suitable for addressing how to use biomaterials serving in resistant infections, antimicrobial therapies and regenerative medicine. It was tested in vivo on mice for MRSA infected wound treatment in mice and pork, and prosthetic joint infection in mice.

Tests have been all clearly discussed as to substantiate the main claim, and are all standing a sharp scrutiny. The results and claims are convincing to me.

The text is clear, to the point and inviting to read. The figures are clear as well.

There is an English typo's grammar mistake that should be corrected. Line 56. In human body should become for example in the (or a) human body.

I have no further comments whatsoever to improve the text or to strengthen the claims made. Therefore, I recommend publish as is.

Reviewer #3

(Remarks to the Author)

In this study, Xuan and colleagues generate an innovative NET "mimic" based on the use of hydrogel composed of lysozyme amyloid flexible. Interestingly, the authors the self-assembled structures can disassemble into unfolded lysozyme monomers, thereby increasing their anti-microbial activity. The authors provide in vitro evidence of low cytotoxicity, high microbicidal properties and in vivo beneficial activity in an MSRA infection skin and bone model, as well as a model of skin infection and wound healing in pigs. The manuscript is clearly presented and well written. However, additional experimental work is necessary to support some of the conclusions stated by the authors.

1. One comment is that the use of Lysozyme as a NET mimic is not in line with the normal composition of these structures. Lysozyme is not a main component of the NET and these structures lack the DNA content and many nuclear proteins such as histones or granule proteins such as MPO, NE and others. Despite structurally resembling NET-like fibers, the definition as NET mimics is not clear to me. The authors should consider renaming and discuss these differences considering the advantages and disadvantages.

2. In line with this previous comment, NETs are known to trap bacteria. The experiments shown don't prove the containment capacity of these structures but rather the microbicidal capacity. Moreover, the authors use MRSA as bacterial model, but this bacterium is not very motile. The authors need to prove that FF contains other bacteria or fungi with more motility and use experimental approaches to prove it. I think is important to consider these structures as NET mimics.

3. Why not FF-ICG has been used in the cytotoxicity tests? The idea of controlling the microbicidal effect by exposing to NIR is interesting but is under these situations when the cytotoxicity of resident cells is affected. Also, employ epithelial or endothelial cells as cell models for these tests to be closer to the MRSA skin model used in vivo.

4. The ability of FF-ICG to induce platelet activation and thrombus formation should be tested. Despite being applied locally, if these structures reach the circulation might cause inflammation and thrombosis.

1. Why do the authors focus on NETs' role in macrophage polarization? NETs typically induce an inflammatory rather than reparative macrophage phenotype. To demonstrate Mg+2's polarizing effect, it should be added at appropriate levels to RF, FFs, or RF-ICGs to assess recovery of CD206 expression.

1. Concerning this question, the observed increase in in vivo CD206 expression and tissue proliferation may not result directly from the induction of a reparative process. It is also possible that enhanced bacterial clearance accelerates the initiation of repair and resolution mechanisms. To clarify this, a wound healing model without bacterial presence could be used to determine whether the application of FF-ICGs results in faster wound closure. This experiment warrants further investigation.

5. Also, the authors should analyze the presence of reparative cytokines in the tissue by ELISA or RNA analysis such as TGFb, IL-10, VEGF or PDGF. Is the supernatant of polarized macrophages inducing epithelial cell proliferation? Also, provide quantification of CD206 and Ki67 of both mouse and porcine models.

6. In the in vivo analysis, ICG treatment already has a strong effect on the survival of the bacteria. What is the effect of the photothermal activation itself? The right comparison wouldn't be FF-ICGs without activation vs FF-ICGs with activation.

1. A relevant question is whether the application offers any benefit when used several days after infection. At 24 hours, biofilm formation is likely minimal, which may enhance treatment effectiveness. If the application occurs several days post-infection, do the authors observe any continued benefit?

7. When considering the potential side-effects of the treatment, the authors should add a cytokine profile in plasma, and leukocyte analysis in blood and tissue, in particular for neutrophils and monocytes.

8. In supplementary figure 27, the heatmap is not clear. If the data is scaled between 0 and 1, there should be some conditions in red. How is this analyzed? Please, the authors should also show in raw counts. Also, add an explanation of the abbreviations in the legend for readability (check for the rest of the legends as well).

Minor:

- The authors should reduce the number of supplementary figures by combining them.

Version 1:

Reviewer comments:

Reviewer #1

(Remarks to the Author)

The authors have carefully and well addressed my comments and suggestions. The article well deserve publication in Nature Communications.

Reviewer #3

(Remarks to the Author)

I greatly appreciate all the efforts made by the authors in addressing all my concerns, and those raised by the other reviewers. The authors have significantly improved the article, and the study is now ready for publication in Nature Communications.

Responses to Reviewers

Photo-reversible Amyloid NanoNETs for Regenerative Antimicrobial Therapies

(Manuscript ID: NCOMMS-25-41589-T)

Responses to Reviewer 1

The authors present an innovative reversible and stimuli-controlled hydrogel with antimicrobial activity ad acting as scaffold for wound healing, with an in-depth in-vitro analysis as well as proof-of-concepts in in-vivo animal models. The study present impressive results on the use of functional protein assembly to design network materials to fight current issue of bacterial resistance to antibiotics. The work is technically sound, and the conclusions are well supported by the presented data. Enough detail is given for the reproducibility of the study. The reviewer suggests publication in Nature Communications, with only the following minor points to be addressed:

Response: We thank the reviewer for positive evaluation of our work. We also appreciate the positive comments and pointed matters. Revisions and corrections have been made accordingly in the revised manuscript.

Comment 1: *Figure 1 is generally clear, but it would benefit from some extra labelling of the panels to describe the molecular mechanisms of action of the hydrogel.*

Response: We thank for the reviewer's kind suggestions. Based on this suggestion, we have revised by adding some more labels to explain the molecular mechanisms and to further improve the readability of **Fig. 1**. The revised **Fig. 1** has been updated as followed:

Editorial Note: Revised Fig.1 in this Peer Review File is reproduced with permission from BioRender, Created in BioRender. Xuan, Q. (2025) <https://BioRender.com/v8dvned>.

Revised Fig.1

Comment 2: In Figure 2b, the authors show that ICG does not significantly affect fibrils morphology, it would be helpful to indicate the number of fibrils analysed and use a statistical test to support further this conclusion.

Response: We thank this constructive comment from the reviewer. Accordingly, we have added the number of fibrils information and the corresponding statistics analysis in the caption of **Figs. 2b-c** (Lines 3-9, Page 9). It reads,

“A large number of fibrils ($n = 449$) were included in the height analysis. The data were expressed as mean \pm standard deviation (S.D). Statistical significance was analyzed by one-way ANOVA using GraphPad Prism 8, followed by Tukey’s post-hoc test for pairwise comparisons. Statistical significance was defined as $*p < 0.05$, $**p < 0.01$, and $***p < 0.001$. Fibril rigidity was studied by measuring the mean square end-to-end distance (MSE2ED) of their polymeric chain³⁷. The morphological data of these fibrils were collectively fitted to obtain

MSE2ED value (c) of FFs/FFs-ICG and RFs/RFs-ICG, with the error bar representing fitting error using the 2D worm-like chain model³⁸.

Comment 3: In Figure 2c is not clear the meaning of the ordinate. The authors should comment also on the origin of the error in the persistence analysis.

Response: We thank for the reviewer's careful reading and thoughtful suggestion. The term MSE2ED on the ordinate denotes the mean square end-to-end distance of the polymeric chain, which is used to estimate the persistence length of these fibrils and consequently their rigidity. These fibrils were collectively fitted to obtain MSE2ED value with the error bar representing fitting error using the 2D worm-like chain model (*Macromolecules* **2015**, 48, 1269-1280).

The corresponding correction has made in the caption of **Figs. 2b-c** of the revised manuscript. It reads,

A large number of fibrils (n = 449) were included in the height analysis. The data were expressed as mean \pm standard deviation (S.D). Statistical significance was analyzed by one-way ANOVA using GraphPad Prism 8, followed by Tukey's post-hoc test for pairwise comparisons. Statistical significance was defined as *p < 0.05, **p < 0.01, and ***p < 0.001. Fibril rigidity was studied by measuring the mean square end-to-end distance (MSE2ED) of their polymeric chain³⁷. The morphological data of these fibrils were collectively fitted to obtain MSE2ED value (c) of FFs/FFs-ICG and RFs/RFs-ICG, with the error bar representing fitting error using the 2D worm-like chain model³⁸.

Comment 4: The authors use in the text the sentence "while in irreversible RFs lysozyme is buried into irreversible aggregates.". While intuitively correct, the reviewer would suggest clarifying that the lysozyme is unfolded to form the fibrillar structure.

Response: We thank for the reviewer's careful correction. We agree with the reviewer and we have revised this sentence (Lines 25-28, Page 10) accordingly as following:

"These results confirm that the nanoconfined bioactive lysozyme in the FFs can be released upon photothermal treatment (Fig. 3e), while irreversible RFs remain intact upon heat treatment due to their stable misfolded β -sheet-rich structure derived from the fully-unfolding state."

Comment 5: Could the authors describe more in detail which is the molecular origin of the disassembly of the FFs fibrils upon thermal treatment?

Response: We thank for the reviewer for this comment. In our previous study (*Nature Communications*, **2024**, 15, 8448), we had investigated the irreversibility of RFs and reversibility

of FFs by cryoEM and solid-state NMR. Our cryoEM study on RFs demonstrated that native lysozyme fully unfolded within 3-h incubation and formed RFs containing 11 inter-molecular in-register β -sheet structure with high thermal stability. In contrast to the well-ordered RFs, our NMR data indicates that the reversible FFs are relatively heterogeneous, composed of a more-ordered intermolecular β -sheet core surrounded by less-ordered helical secondary structural elements reminiscent of a molten globule-like state. This indicate that the core of the reversible FFs, formed only 5 min of unfolding and fibrillization, is likely to consist primarily of the partially unfolded β -sheet region of the globular lysozyme, yielding metastable fibril-forming β -sheet segments. From energy landscape's perspective (see figure below), these metastable β -sheet segment in the fibril should remain a relatively high energy state and short energy barrier between native lysozyme. Thus, these FFs can break and degrade into monomeric lysozyme upon heat treatment, inducing the disassembling of FFs. More details can be seen here: <https://doi.org/10.1038/s41467-024-52681-z>

[Figure Redacted]

Energy landscape of FF and RF, adapted from Figure 5 in our previous study (*Nature Communications*, 2024, 15, 8448)

Comment 6: Which NIR power was used in the *in-vivo* experiments? How much in depth could go for further extension of this powerful approach?

Response: We thank the reviewer for his/her careful reading. In the *in-vivo* experiments, we used an 808 nm NIR at a power density of 0.3 W/cm² with an irradiation duration of 5 min for both murine wound and periprosthetic joint infection models, and at a power density of 0.5 W/cm² with an irradiation duration of 10 min for porcine wound model. The slightly higher power and longer duration for the porcine model were chosen to account for the greater thickness and scattering properties of porcine skin to ensure sufficient energy reached the hydrogel. These NIR parameters were carefully optimized to avoid thermal damage and ensure enhanced clinical safety (**Supplementary Fig. 6**). More importantly, this specific parameter selection allowed us to accurately evaluate the inherent reversible properties of FFs, rather than relying solely on photothermal ablation for bacterial elimination. We have added these key details of NIR parameters in the section of Methods in the revised manuscript.

The reviewer raises a critical point regarding the depth limitations of NIR light, which is a key consideration for translational applications. The penetration depth of 808 nm light in biological tissue is typically limited to a few millimeters to a centimeter, largely depending on the tissue type and composition. This well-established penetration capability has been

previously documented (*Nature Biotechnology* **2004**, *22*, 969-976; *Nature Biotechnology* **2001**, *19*, 316-317; *Chemical Review* **2017**, *117*, 3, 901-986). Our current study demonstrates the high efficacy of this platform for treating surface and shallow tissue infections, such as skin wounds and peri-implant infections adjacent to the joint space, which are perfectly suited for this depth of penetration. The power of this approach can be extended to deeper-seated infections through the following two promising strategies. (i) Alternative light delivery systems: The use of fiber-optic probes or implantable NIR devices could be integrated to deliver NIR light precisely to deeper tissues, a technique already being explored in other photothermal and photodynamic therapies (*Advanced Optical Materials* **2024**, *12*, 2400478; *Cancers* **2021**, *13*, 3484). (ii) Material optimization: Adjusting the photothermal agent (e.g., using agents with higher photothermal conversion efficiency at longer NIR wavelengths, such as in the second biological window NIR-II) could allow for the use of lower power densities and achieve greater depth penetration with reduced scattering (*Nature Photonics* **2024**, *18*, 535-547). In conclusion, while the penetration depth of NIR light is an inherent physical limitation, the flexibility of our biomaterial platform and the feasibility of complementary clinical techniques make it a powerful and adaptable strategy. Its potential extends beyond superficial wounds to a range of localized bacterial infections accessible *via* direct application or minimally invasive procedures.

Responses to Reviewer 2

The article presents a highly innovative approach for applying anti-microbial therapies by means of using flexible amyloid fibers that formed reversibly. Then innovative power lies in the combination of choosing the right protein that is forming amyloid fibers reversibly. In other words: the authors have found a route to mitigate cytotoxicity, while preserving intrinsic antibacterial activity of the proteins added. Even more specialistic, the flexible fibers play two roles, one of entrapment of bacterial species that do harm, and the degradation by the proteins forming reversibly these flexible fibers. I think this is a truly amazing and very unexpected finding. The authors combine dynamic responsiveness, multifunctionality and biocompatibility.

Amazingly, although a heat shock treatment is taking place to induce fiber formation, the fiber formation is reversible, and the proteins remain enzymatically active. The reversibility allows for photo-induction of disintegration and subsequent efficacy. The impact of the work is large as it is suitable for addressing how to use biomaterials serving in resistant infections, antimicrobial therapies and regenerative medicine. It was tested in vivo on mice for MRSA infected wound treatment in mice and pork, and prosthetic joint infection in mice. Tests have been all clearly discussed as to substantiate the main claim, and are all standing a sharp scrutiny. The results and claims are convincing to me. The text is clear, to the point and inviting to read. The figures are clear as well. There is an English typo's grammar mistake that should be corrected. Line 56. In human body should become for example in the (or a) human body. I have no further comments whatsoever to improve the text or to strengthen the claims made. Therefore, I recommend publish as is.

Response: We sincerely appreciate the reviewer's recognition of this work firstly. We sincerely thank the reviewer for his/her recommendation for publication. We also thank the reviewer for their careful reading and pointed grammar mistake. We have corrected this phrase from "In human body" to "In the human body" as suggested, which have been incorporated into the revised manuscript (Line 13, Page 3).

Responses to Reviewer 3

In this study, Xuan and colleagues generate an innovative NET “mimic” based on the use of hydrogel composed of lysozyme amyloid flexible. Interestingly, the authors the self-assembled structures can disassemble into unfolded lysozyme monomers, thereby increasing their anti-microbial activity. The authors provide in-vitro evidence of low cytotoxicity, high microbicidal properties and in-vivo beneficial activity in an MSRA infection skin and bone model, as well as a model of skin infection and wound healing in pigs. The manuscript is clearly presented and well written. However, additional experimental work is necessary to support some of the conclusions stated by the authors.

Response: We sincerely appreciate the reviewer’s constructive comments and suggestions, which have significantly improved the quality of our manuscript. Accordingly, we provide point-by-point responses to the raised concerns, with revisions highlighted in the revised manuscript.

Comment 1: *One comment is that the use of Lysozyme as a NET mimic is not in line with the normal composition of these structures. Lysozyme is not a main component of the NET and these structures lack the DNA content and many nuclear proteins such as histones or granule proteins such as MPO, NE and others. Despite structurally resembling NET-like fibers, the definition as NET mimics is not clear to me. The authors should consider renaming and discuss these differences considering the advantages and disadvantages.*

Response: We appreciate and respect the reviewer’s critical insight into the terminology of “neutrophil extracellular traps (NETs)” from the perspective of structural composition. We totally agree that native NETs are primarily composed of DNA, histones, myeloperoxidase (MPO), neutrophil elastase (NE), and other granular proteins, whereas lysozyme is not a major component (*Cardiovascular Research*, **2022**, 118, 2737-2753; *Nature Review Immunology*, **2018**, 18, 134-147). Our intention was not to fully replicate the biochemical composition of NETs, but rather to mimic their functional behavior, especially their ability to entrap and neutralize pathogens through a fibrous network. Lysozyme, as a naturally-occurring host defense peptide and neutrophil-derived antimicrobial granule protein, shares functional similarities with certain NET components (e.g., MPO and NE) in its ability to disrupt bacterial cell integrity and participates in the killing of bacteria trapped in the NETs. (*PLoS Pathogen*, **2017**, 13, e1006512; *Blood*, **2005**, 106, 2551-2558; *Annual Review of Immunology*, **2005**, 23, 197-223). Additionally, this concept of “functional mimicry of NETs” is well-established in the biomaterials-related studies, where synthetic systems are designed to capture the essential biological function of a natural structure while offering advantages such as simplified composition, tunability, and improved biocompatibility (*Nature Communications*, **2024**, 15, 555; *ACS Nano*, **2025**, 19, 13, 13202-13219). For instance, amphiphilic peptide molecules have

already been employed to self-assemble into nanofibrous structures resembling NETs networks, effectively trapping and killing methicillin-resistant *Staphylococcus epidermidis* (*Biomaterials*, 2020, 253: 120124). Another work designed positively charged nanoparticles that emulate the “capture-and-kill” mechanism of NETs by entrapping negatively charged bacteria and eliminating them through photothermal effects (*Biomaterials Science*, 2024, 12, 1841-1846). We hope this clarification addresses the reviewer's concern and demonstrates that our naming is consistent with the established paradigm in the field.

In response to the reviewer's suggestion, we have added a dedicated clarification in the section of Introduction (Lines 8-17, Page 5) to explicitly compare and contrast our system with native NETs. It reads,

“Together, our findings establish a proof-of-concept for NETs-inspired materials based on biocompatible HDPs with controllable enzymatic activity. It is noted that our biomimetic engineering strategy focuses on functional emulation rather than structural replication, aiming to create a minimalist, HDP-based functional mimic that achieves the sequential “capture-and-kill” dynamic of NETs. The nanofibrous network acts as a physical trap for bacteria “capture”, analogous to the DNA/protein fibers of native NETs. The photothermally-triggered release of bioactive lysozyme provides the antimicrobial “kill” function, analogous to the role of MPO, NE, and other components in native NETs. This study opens promising avenues for the development of next-generation biomaterials serving in resistant infections, antimicrobial therapies and regenerative medicine.”

We thank for the reviewer to provide us a chance to elaborate this biomimetic concept again. We believe these revisions will provide a more precise and scientifically rigorous framing of our work, while still highlighting its innovative nature as a NETs-inspired therapeutic platform.

Comment 2: In line with this previous comment, NETs are known to trap bacteria. The experiments shown don't prove the containment capacity of these structures but rather the microbicidal capacity. Moreover, the authors use MRSA as bacterial model, but this bacterium is not very motile. The authors need to prove that FF contains other bacteria or fungi with more motility and use experimental approaches to prove it. I think is important to consider these structures as NET mimics.

Response: The authors thank for this insightful and constructive comment, which rightly emphasizes the importance of demonstrating the containment or trapping capacity of our nanoNETs, a defining feature of native NETs. The initial data in our original manuscript provided some preliminary evidence for bacterial containment. For instance, Supplementary Fig. 18 showed that our FFs possess a highly positive surface charge, which facilitates strong

electrostatic interactions with the negatively charged surfaces of most bacteria. SEM images in **Fig. 3h** visually showed MRSA cells adhered to the fibrillar network of FFs.

In direct response to the reviewer's suggestion, we further evaluated the pathogen capture and antimicrobial efficacy of our nanoNETs against *Pseudomonas aeruginosa* (*P. aeruginosa*, a highly motile Gram-negative bacteria known for its swimming and swarming motility) (*Annual Review of Microbiology*, 2012, 66, 493-520), also involved *Candida albicans* (*C. albicans*, a fungal strain) to further demonstrate its broad-spectrum potential beyond bacteria. As shown in Supplementary Fig. 17, our nanoNETs can achieve the effective physical entrapment of both MRSA and *P. aeruginosa* within the fibrous network, visually confirmed by SEM images. These images offer direct morphological evidence of bacterial capture, regardless of bacterial motility. We have also experimentally confirmed that both strains (MRSA and *P. aeruginosa*) exhibit a negative surface charge, as measured by zeta potential analysis in Supplementary Fig. 18. This characteristic enables them to be effectively captured through electrostatic adsorption by the positively-charged nanoNETs. We also verified the antibacterial efficacy of nanoNETs against MRSA, *P. aeruginosa*, and *C. albicans*. Based on the CFU counting results (Supplementary Fig. 22), the bacterial survival rates were significantly reduced by 95.4±0.6% for MRSA, 86.2±4.3% for *P. aeruginosa*, and 89.8±4.7% for *C. albicans* after treatment with our nanoNETs. In comparison, the nanoNETs displays the highest antibacterial efficiency against Gram-positive bacteria (MRSA), largely attributable to the action mechanism of lysozyme itself (*Antibiotics*, 2021, 10, 1534). Meanwhile, the nanoNETs also shows effective antibacterial activity against Gram-negative bacteria and fungi, demonstrating its broad-spectrum antibacterial activity. These results collectively provide direct evidence supporting the “trap-and-kill” mechanism of the nanoNETs.

These new results have been presented in Supplementary Figs. 17-18 and Fig. 22 of **Supplementary Information** file, and the corresponding discussion has been given in the section of Results and discussion (Lines1-25, Page 11) of revised manuscript. It reads,

“To demonstrate the pathogen-trapping and killing capacity of our nanoNETs, we utilized MRSA and *Pseudomonas aeruginosa* (*P. aeruginosa*), a highly motile Gram-negative bacteria known for its swimming and swarming motility) as model strains. As shown in Supplementary Fig. 17, our nanoNETs can achieve the effective physical entrapment of both MRSA and *P. aeruginosa* within the fibrous network, visually confirmed by scanning electron microscope (SEM) images, which offer direct morphological evidence of bacterial capture, regardless of bacterial motility. We have also experimentally confirmed that both strains (MRSA and *P. aeruginosa*) exhibit negative surface charges, as measured by zeta potential in Supplementary Fig. 18. This characteristic enables them to be effectively captured through electrostatic adsorption by the positively charged nanoNETs^{29,40}. We found that the lysozyme monomers exhibited a high antimicrobial efficiency (89.2±2.5%), outperforming RFs (62.1±5.5%) and

FFs (71.9±0.7%) (Fig. 3f and Supplementary Fig. 19). RFs-ICG achieved a higher antibacterial rate (81.5%±3.8%) partially increased by ICG, whereas FFs-ICG demonstrated the highest antibacterial performance (95.4%±0.6%) within low-power NIR irradiation (0.3 W/cm²). Notably, this low photothermal power allows to accurately assess the differences in antibacterial activity between RFs and FFs. These results are supported by the live/dead bacterial staining images (Fig. 3g). Further, SEM imaging (Fig. 3h) shows the MRSA membrane disruption, and the leakage of N-acetyl-beta-D-glucosidase (NAG) and K⁺ after treatments (Supplementary Fig. 20-21), indicating the increased bacterial membrane permeability as the antibacterial activity. Additionally, FFs-ICG nanoNETs exhibited an increased antimicrobial efficacy of 86.2±4.3% for *P. aeruginosa*, and 89.8±4.7% for *Candida albicans* (a typical fungal strain) (Supplementary Fig. 22), demonstrating the broad-spectrum antibacterial efficacy of nanoNETs beyond Gram-positive bacteria. These results indicated that our FFs possess a synergistic effect against bacteria: NIR-induced FFs partially disassemble and released active lysozyme for bacteria degradation, while the residual FFs-ICG nanoNETs maintain the function of pathogen entrapment through the absorption-contact mechanism as conventional RFs^{41,42}, collectively providing direct evidence to support the “trap-and-kill” mechanism of the nanoNETs.”

We believe these supplementary results can unequivocally demonstrate the pathogen-trapping and killing capacity of our nanoNETs, solidifying its functional analogy to native NETs despite the compositional differences highlighted in *Comment 1*. We are grateful for this suggestion from the reviewer, which significantly strengthens the functional claim of our work.

Supplementary Fig. 17 (Note: red arrows represent the FFs networks)

Supplementary Fig. 18

Supplementary Fig. 22

Comment 3: Why not FF-ICG has been used in the cytotoxicity tests? The idea of controlling the microbicidal effect by exposing to NIR is interesting but is under these situations when the cytotoxicity of resident cells is affected. Also, employ epithelial or endothelial cells as cell models for these tests to be closer to the MRSA skin model used in vivo.

Response: We thank the reviewer for raising this important point. The primary purpose of the cytocompatibility assays presented in **Fig. 3a** and Supplementary Figs. 12-13 was to evaluate the inherent biocompatibility of the biomaterials themselves (lysozyme monomers, FFs, RFs), independent of their therapeutic function. This approach is standard practice for initial biomaterial screening (*Advanced Functional Materials* **2025**, 2501242; *Advanced*

Functional Materials **2024**, *34*, 2402531; *ACS Nano* **2024**, *18*, 26961-26974), as it isolates the material's chemical properties from external physical stimuli (like heat) that are part of its intended application mechanism. We agree that the local temperature increase during photothermal process can indeed affect cell viability. However, this effect is an intended therapeutic outcome targeting pathogens, not an intrinsic toxicity of the materials. The key distinction we aim to demonstrate is that our nanoNETs are inherently benign and that the antimicrobial effects are selectively activated by NIR. Furthermore, it is important to clarify that the *in-vivo* photothermal treatment is only applied topically and transiently (0.3 W/cm², 5 min for murine models and 0.5 W/cm², 10 min for porcine model), causing a localized and temporary temperature increase that is well-tolerated by tissues (~40°C for murine model and ~50°C for porcine model) due to its buffering capacity and blood perfusion (**Supplementary Fig. 6**). Moreover, such photothermal temperature belongs to the reported safe temperature ranges (*Science Advance*, **2020**, *6*, eabb1311; *Advanced Functional Materials*, **2021**, *31*, 2100738), and the photothermally-triggered release of key components, including lysozyme monomers, Mg²⁺, and ICG (FDA-approved drug for thallium detoxification) does not cause significant cytotoxicity to resident skin cells. The biocompatibility of these components under experiment conditions in this study has been proved by previous studies (*Chemical Engineering Journal*, **2020**, *396*, 125335; *ACS Nano* **2023**, *17*, 23498). Therefore, while a dedicated *in-vitro* cytotoxicity assay under NIR could be performed, we believe the combined *in-vivo* evidence (**Figs. 4e-h** and Supplementary Figs. 28-29), demonstrating excellent tissue regeneration and a complete lack of systemic toxicity, offers a more physiologically relevant and conclusive assessment of the treatment's safety profile.

Additionally, we have performed cytotoxicity assays using human umbilical vein endothelial cells (HUVECs) as suggested by the reviewer. The results (Supplementary Fig. 14) are consistent with those from L-929 cells, and the cell viabilities of all components are exceeding 80%, with 1 wt.% FFs-ICG (nanoNETs) showing the highest cell viability of 95.0±8.1%.

We have added these data and the corresponding explanation to the section of Results and Discussion (**Lines 6-13, Page 10**) in the revised manuscript. It now reads:

“To confirm this, we first applied L-929 fibroblast in the assessment of FFs cytocompatibility (Fig. 3a). Results indicate that lysozyme monomers (1 wt.%) reduced cell viability to 50 %, whereas both RFs and FFs maintain cell survival to 90%. This suggests that the linear alignment of lysozyme significantly mitigates the cytotoxic effect of native lysozyme. Live/dead-staining confocal imaging results verified the cytocompatibility of the FFs hydrogel (Supplementary Fig. 12). The concentration-dependent cytotoxicity in L-929 cells was also evaluated (Supplementary Fig. 13), demonstrating an excellent biocompatibility of the FFs hydrogel, even at high-concentration exceeding 1 wt.%. Similar results were observed in the

cytotoxicity assays using human umbilical vein endothelial cells (HUVECs) (Supplementary Fig. 14).”

We thank the reviewer for raising this important point again. Our new data robustly demonstrate that FF-ICG exhibits excellent cytocompatibility in both fibroblast and skin-relevant HUVECs, further supporting its potential for safe and effective use in antimicrobial and regenerative therapies.

Supplementary Fig. 14

Comment 4: *The ability of FF-ICG to induce platelet activation and thrombus formation should be tested. Despite being applied locally, if these structures reach the circulation might cause inflammation and thrombosis.*

Response: We are grateful to the reviewer for this crucial suggestion regarding the potential thrombogenic risk of nanoNETs. We fully agree that assessing any possible platelet activation or thrombotic risk is essential for evaluating the biosafety of biomaterials, especially those with fibrillar structures, even for topical applications, due to the potential for unintended systemic exposure through compromised vasculature in wounds. For this purpose, we fabricated the MRSA-infected murine wound model again and performed two different treatments, including PBS (control) and FFs-ICG (nanoNETs). After that, the blood samples of treated mice were collected and analyzed *in-vitro*. The coagulability of FFs-ICG was first assessed by measuring the blood coagulation index (BCI) according to a previous study (Small, 2025, 2503490), which demonstrated the overall effect of the materials on blood coagulation. It is noted that the thrombin-treated blood samples from the PBS group serve as the positive control. Thrombin is a multifunctional serine protease in the coagulation cascade, and plays a central role in the functioning of hemostasis, with the function of cleaving fibrinogen to fibrin, which forms the fibrin clot of a hemostatic plug (Blood Coagulation & Fibrinolysis 2022, 33, 145-148). As shown in Supplementary Fig. 30a, the BCI of the FFs-ICG group is very close to that of the PBS group, significantly larger than to that of the thrombin group, indicating that FFs-ICG does not have significant impact in promoting blood coagulation. We further analyzed

platelet activation conditions using the double staining of CD61 (platelet marker) (*Platelets* 2021, 32, 786-793) and CD62P (P-selectin, a key marker of platelet activation) (*Clinical Experimental HEPATOLOGY* 2021, 7, 231-240) by flow cytometry. Notably, the elevated CD62P expression is a well-established marker of platelet activation and increased risk of thrombotic diseases (*Clinical Experimental HEPATOLOGY* 2021, 7, 231-240). As shown in Supplementary Figs. 30b-c, the percentage of CD61⁺/CD62P⁺ platelets maintained a normal level and comparable to the PBS control, while the thrombin treatment significantly increased the proportion of activated platelets, increasing the risks of thrombus formation. These comprehensive data strongly indicate that FFs-ICG nanoNETs possess a low risk of inducing thrombosis, supporting their safety for proposed topical applications. The lack of thrombogenicity is likely due to the biocompatible nature of lysozyme and the flexible, non-rigid structure of the FFs, which is less prone to mechanically triggering coagulation cascades compared to stiff fibrous materials.

We have added these new data in Supplementary Figs. 30 of **Supplementary Information** file, and the corresponding explanation to the section of Results and Discussion (Lines 14-19, Page 15) in the revised manuscript. It reads,

“More importantly, our FFs-ICG nanoNETs also exhibited a low risk of inducing thrombosis (including platelet activation and thrombus formation), supporting their safety for proposed topical applications (Supplementary Fig. 30). The lack of thrombogenicity is likely due to the biocompatible nature of lysozyme and the flexible, non-rigid structure of the FFs, which is less prone to mechanically triggering coagulation cascades compared to stiff fibrous materials.”

Supplementary Figs. 30

Comment 5: Why do the authors focus on NETs' role in macrophage polarization? NETs typically induce an inflammatory rather than reparative macrophage phenotype. To demonstrate Mg²⁺'s polarizing effect, it should be added at appropriate levels to RF, FFs, or RF-ICGs to assess recovery of CD206 expression.

Response: We thank the reviewer for this insightful comment, which allows us to clarify a

critical aspect of our biomimetic strategy and to provide definitive evidence for the role of Mg^{2+} in macrophage polarization. The reviewer is absolutely correct that native NETs, particularly in their pathological context, are often associated with sustained inflammation and can impede resolution. Our aim was not to replicate the immunostimulatory profile of NETs, but rather their core physical “trap-and-kill” function. We explicitly state in the Introduction that “dysregulated or excessive NETs release can lead to tissue damage and chronic inflammation,” and our goal was to create a synthetic mimic that avoids these detrimental effects while harnessing the beneficial antimicrobial mechanism.

Our focus on macrophage polarization stems not from the NETs-mimetic structure itself, but from the unique functional outcome of its stimuli-responsive disassembly: the release of Mg^{2+} . We hypothesize that the true regenerative power of our platform lies in this subsequent, controlled release of a pro-healing agent after the antimicrobial action is complete. This sequential functionality (first antimicrobial, then pro-regenerative) is a key innovative aspect of our design, distinguishing it from static NETs mimics.

Regarding the concern about Mg^{2+} concentration consistency across samples, we would like to clarify that all hydrogel samples for macrophage polarization regulation are prepared using the equal Mg^{2+} concentration. Prior to evaluation, these hydrogel samples were immersed and treated (with or without NIR irradiation) to acquire aqueous extracts, which were then added into the medium and co-incubated with macrophage. Due to the photothermal-responsive characteristics, the extracts of FFs-ICG nanoNETs indeed contained higher Mg^{2+} concentration than any other groups (RFs, FFs, and RFs-ICG), which has been confirmed by the release curves (**Fig. 3i**). This treatment precisely mimics the actual situation of Mg^{2+} released by the hydrogel in the body, and therefore it can better reflect the regulatory ability of macrophage polarization. The ultimate proportion of M2-type macrophage ($CD206^+$) was also consistent with the released Mg^{2+} concentration, provide definitive evidence for the role of Mg^{2+} in macrophage polarization.

We have provided more detailed method in the section of the Macrophage polarization assays in **Supplementary Information** file (Page 7), and the corresponding explanation in Results and Discussion (Lines 2-10, Page 12) in the revised manuscript. It reads,

“Mouse-derived macrophage (Raw246.7) was utilized to test magnesium ions (Mg^{2+})-mediated macrophage polarization towards M2 phenotype according to previous studies^{7, 8}. Specifically, 1×10^5 cells/well were cultured in DMEM medium including 10% fetal bovine serum (FBS) and 1% penicillin/streptomycin in 96-well plates at 37°C in a humidified atmosphere of 5% CO_2 . After 12-hour incubations, the medium in cell cultures was replaced by the FBS-free medium. Meanwhile, different samples (PBS, IL-4, RFs, FFs, RFs-ICG, and FFs-ICG) were prepared as hydrogel, then immersed in 1 mL of FBS-free medium. RFs-ICG and FFs-ICG groups were treated using 808 nm laser irradiation for 10 min. Subsequently, these

samples were filtered by using a 0.22 µm filter membrane, and then the Mg²⁺-containing extract solution was added to the above cell cultures and incubated for 1 day. The cells were collected by discarding the old medium, and each well was fixed by adding 80% methanol solution for 40 min at 4°C. After centrifugation, the excess solution was removed, and the fresh DMEM containing 5 µL of CD206 antibody with red fluorescence was added and incubated at 37°C for 60 min. Next, the superfluous unbinding dyes were removed prior to macrophage polarization analysis using flow cytometry. For CLSM imaging, the treated cells were first fixed with the addition of 80% methanol solution at 4°C for 30 min, after which the excess solution was removed and fresh DMEM containing 5 µL of CD206 antibody with red fluorescence was added to stain the samples for 60 min. After the excess dye was removed, the polarization status of cells was observed and recorded using CLSM.” (Supplementary Information)

“To confirm this, we used Raw 264.7 cells as the macrophage model to assess the immunomodulatory capacity of our nanoNETs, with IL-4 as the positive control for M2 polarization. We observe a clear correlation between Mg²⁺ release and M2-type polarization (Figs. 3j-1), with the highest M2 polarization ratio (36.7±1.3%) in FFs-ICG group due to the photothermal-responsive characteristics. Dysregulated or excessive native NETs release can lead to tissue damage and chronic inflammation, whereas our nanoNETs can avoid these detrimental effects and harnessing the beneficial antimicrobial mechanism. The data above provide definitive evidence for the role of released Mg²⁺ from nanoNETs in macrophage polarization to prevent the occurrence of these adverse outcomes.” (Lines 2-10, Page 12 in Manuscript)

Comment 6: Concerning this question, the observed increase in in vivo CD206 expression and tissue proliferation may not result directly from the induction of a reparative process. It is also possible that enhanced bacterial clearance accelerates the initiation of repair and resolution mechanisms. To clarify this, a wound healing model without bacterial presence could be used to determine whether the application of FF-ICGs results in faster wound closure. This experiment warrants further investigation.

Response: We thank the reviewer for this constructive comment, which compels us to rigorously distinguish between the direct pro-regenerative effects of our nanoNETs and the indirect benefits secondary to their antimicrobial action. We agree that this is a fundamental question for establishing the multifunctional nature of our platform. To directly address this, we conducted a new *in-vivo* experiment using a sterile (non-infected) murine full-thickness wound healing model. The experimental design mirrored our infected model, but without MRSA inoculation. Wounds were treated with PBS and FFs-ICG (+NIR), and wound closure was monitored over 10 days. As shown in Supplementary Figs. 31a-c, the FFs-ICG group demonstrated significantly faster wound closure kinetics, with a reduction of 8.7% in wound area, compared to the control group in the sterile

environment. Histological (H&E) analysis at day 10 revealed that the FFs-ICG group exhibited thicker, more organized granulation tissue and advanced re-epithelialization compared to the control. Immunohistochemical detection of FFs-ICG-treated group revealed a 1.7-fold increase in Ki-67 and 2.2-fold rise in IL-10 compared to the control group (Supplementary Figs. 31f-i), suggesting the enhanced cellular proliferation potential and increased anti-inflammatory activity. Furthermore, immunofluorescence analysis revealed the significant increase of CD206 expression in the group of FFs-ICG, with a 1.9-fold enhancement over the control group (Supplementary Fig. 31j-k). Collectively, these supplementary data confirm that FFs-ICG nanoNETs platform possesses intrinsic pro-regenerative properties that operate independently of its antimicrobial function.

The results of these data are now included as Supplementary Fig. 31 and discussed in the section of Results and Discussion (Lines 8-11, Page 16) of the revised manuscript. It reads,

“This hypothesis was further verified by the results from a sterile (non-infected) murine full-thickness wound healing model (Supplementary Fig. 31), which confirmed that FFs-ICG nanoNETs platform possessed intrinsic pro-regenerative properties that operate independently of its antimicrobial function.”

Supplementary Fig. 31

Comment 7: Also, the authors should analyze the presence of reparative cytokines in the tissue by ELISA or RNA analysis such as TGF- β , IL-10, VEGF or PDGF. Is the supernatant of polarized macrophages inducing epithelial cell proliferation? Also, provide quantification of CD206 and Ki67 of both mouse and porcine models.

Response: We sincerely thank the reviewer for these insightful suggestions (from *Comment 5* to 7), which have significantly strengthened the mechanistic elucidation in the regenerative properties of our nanoNETs. As suggested by the reviewer, we further performed quantitative PCR validation on wound tissues and observed significant upregulation in the mRNA expression of the genes related to key reparative cytokines and growth factors in the FFs-ICG group, including *Vegfa*, *Hif1 α* , *Il10*, and *Pdgfb*, compared to the PBS control (Supplementary Fig. 26). This provides direct molecular evidence that our treatment actively promotes a pro-regenerative transcriptional program within the wound microenvironment, perfectly aligning with the observed functional outcomes of accelerated healing. In detail, the increase in *Vegfa* (VEGFA) and *Pdgfb* (PDGF-b) indicates enhanced angiogenesis and fibroblast proliferation (*Journal of Controlled Release*, 2024, 373, 319-335), while *Hif1 α* (HIF-1 α) upregulation indicates enhanced angiogenesis and reduced inflammation (*Journal of Controlled Release*, 2025, 380, 330-347). Furthermore, the upregulation of *Il10* (IL-10) confirms a shift toward an anti-inflammatory and pro-regenerative immune microenvironment (*Nature Communications*, 2024, 15, 2939).

Supplementary Fig. 26

In responses to the comment “*Is the supernatant of polarized macrophages inducing epithelial cell proliferation?*”, we collected supernatant from macrophages treated with FFs-ICG (with NIR irradiation), which was then added into the medium and co-incubated with L-929 cells. The result revealed that this supernatant significantly enhanced L-929 cells proliferation by 39.9% compared to the control group (Supplementary Fig. 23), further supporting the presence of reparative cytokines in the supernatant and highlighting the functional role of M2-polarized macrophages in promoting cellular growth and tissue regeneration.

Supplementary Fig. 23

In response to the comment “Also, provide quantification of CD206 and Ki67 of both mouse and porcine models”, we have provided quantitative analysis of immunohistochemical staining for Ki-67 and immunofluorescent staining for CD206 by ImageJ in mouse (Supplementary Figs. 25c, f) and porcine wound models as followed (Fig. 6k and Supplementary Fig. 33f).

Supplementary Figs. 25c, f

Supplementary Fig. 33f

The results of these data are now included in **Supplementary Information** and discussed in the section of Results and Discussion of the revised manuscript. It reads,

“We further performed quantitative PCR validation on wound tissues and observed significant upregulation in the mRNA expression of the genes related to key reparative cytokines and growth factors in the FFs-ICG group, including *Vegfa*, *Hif1 α* , *Il10*, and *Pdgfb*, compared to the PBS control (Supplementary Fig. 26). This provides direct molecular evidence that FFs-ICG actively promotes a pro-regenerative transcriptional program within the wound microenvironment, perfectly aligning with the observed functional outcomes of accelerated healing.” (Lines 28-30, Page 14; Lines 1-3, Page 15)

“Additionally, we collected supernatant from macrophages treated with FFs-ICG, which was then added into the medium and co-incubated with L-929 cells. The result revealed that this supernatant significantly enhanced L-929 cells proliferation by 39.91% compared to the control group (Supplementary Fig. 23), suggesting the presence of reparative cytokines in the supernatant and thus highlighting the functional role of M2-polarized macrophages in promoting cellular growth and tissue regeneration.” (Lines 10-15, Page 12)

Comment 8: In the in vivo analysis, ICG treatment already has a strong effect on the survival of the bacteria. What is the effect of the photothermal activation itself? The right comparison would not be FF-ICGs without activation vs FF-ICGs with activation.

Response: We thank the reviewer for his/her careful reading and insightful comment. We apologize for the fuzzy annotations (all groups incorporating ICG were performed in the presence of NIR irradiation in this study, e.g. ICG, RFs-ICG, and FFs-ICG), which may have caused misunderstandings from the reviewer. Additionally, it is noted that indocyanine green (ICG), as a typical photothermal agent, can only transform NIR to local heating, enabling the occurrence of photothermal conversion in this study. However, ICG molecule itself has no any antibacterial activity without NIR irradiation (*ACS Omega* **2022**, *7*, 38, 33821-33829; *ACS Applied Polymer Materials* **2025**, DOI: 10.1021/acsapm.5c01387). In addition, our present results verified that ICG+NIR can only achieve an excellent antibacterial efficiency but less in tissue regeneration and inflammation elimination in MRSA-infected murine wound model (**Fig. 4**). While in MRSA-infected periprosthetic joint infection murine model, its advantages of antimicrobial performance also significantly diminished, not to mention its role in bone tissue repair and the restoration of walking ability (**Fig. 5**). Consequently, we have set up different treatments in the *in-vivo* model to distinguish the contribution of photothermal effects (FFs vs. FFs-ICG+NIR) and the release of bioactive lysozyme/Mg²⁺ (ICG+NIR vs. FFs-ICG+NIR). Our results confirmed that the photothermal effect (ICG+NIR) provides a strong, non-specific antibacterial action, while the FFs without NIR provides a passive “trap” function and a

biocompatible scaffold. The synergistic effect was achieved in FFs-ICG+NIR, and the photothermal effect triggers the on-demand release of the specific biological “kill” agent (lysozyme) and the pro-healing agent (Mg^{2+}), while the fibrous network provides a protective scaffold that mitigates the potential collateral damage of pure PTT and supports cell growth. It is the photothermal (ICG)-controlled dynamic functionality of the nanoNETs, the ability to convert from a passive trap to an active antimicrobial and regenerative factory, that delivers the optimal therapeutic outcome.

We have added the specific notes in the caption of Figures with ICG annotations, and amended the text in the Results and Discussion sections (Line 29, Page 15; Lines 1-8, Page 16) to emphasize this point and to frame our conclusions around this more rigorous comparison. It reads,

“Additionally, our results facilitate to distinguish the contribution of photothermal effects and the release of bioactive lysozyme/ Mg^{2+} . The photothermal effect (from ICG) provides a strong, non-specific antibacterial action, while the FFs provides a passive “trap” function and a biocompatible scaffold. The synergistic effect was achieved in FFs-ICG+NIR, where the photothermal effect triggers the on-demand release of the specific biological “kill” agent (lysozyme) and the pro-healing agent (Mg^{2+}), while the fibrous network provides a protective scaffold that mitigates the potential collateral damage of pure PTT and supports cell growth. It is the photothermal (ICG)-controlled dynamic functionality of the nanoNETs, the ability to convert from a passive trap to an active antimicrobial and regenerative factory, that delivers the optimal therapeutic outcome.”

***Comment 9:** A relevant question is whether the application offers any benefit when used several days after infection. At 24 hours, biofilm formation is likely minimal, which may enhance treatment effectiveness. If the application occurs several days post-infection, do the authors observe any continued benefit?*

Response: We thank the reviewer for raising this critically important point, which directly addresses the translational potential of our therapy for treating established infections, a common scenario in clinical practice. To directly answer this question, we conducted a new set of experiments in our MRSA-infected murine wound model where the first treatment was delayed until 72 hours post-infection. This time point allows for robust biofilm formation, as confirmed in previous studies (*Chemical Engineering Journal* **2025**, 508, 160905; *ACS Nano* **2025**, 19, 10922-10942). Our results proved that FFs-ICG nanoNETs still could significantly accelerate the wound healing process with a 15.7% decrease in wound area (Supplementary Figs. 27 a-c), despite mature biofilm formed. Hematoxylin & eosin (H&E) staining analysis (Supplementary Fig. 27 d) further demonstrated accelerated regeneration in the FFs-ICG group compared to the control group, characterized by complete dermis and epidermis reconstruction, smaller wound size, and nascent hair follicle formation. Giemsa staining confirmed the excellent antimicrobial effect of the FFs-ICG

nanoNETs (Supplementary Fig. 27 e). Immunohistochemical detection of FFs-ICG nanoNETs-treated group revealed a 2.3-fold increase in Ki-67 and 2.5-fold rise in IL-10 compared to the control group (Supplementary Figs. 27f-i), suggesting the enhanced cellular proliferation potential and increased anti-inflammatory activity. Furthermore, immunofluorescence analysis revealed the significant increase of CD206 expression in the group of FFs-ICG, with a 2.8-fold enhancement over the control group (Supplementary Figs. 27j-k). Collectively, these supplementary data confirm that the therapeutic benefit of our FFs-ICG nanoNETs extends to established infections with mature biofilms. The photothermally-activated release of bioactive lysozyme provides a decisive advantage in this challenging scenario by enabling direct enzymatic disruption and killing of biofilm-embedded bacteria, which is not possible with static, non-responsive biomimetic NETs. This significantly enhances the clinical relevance of our platform, as it demonstrates efficacy not just as a prophylactic or early intervention, but as a potential treatment for ongoing, resistant infections.

These new results have been presented in **Supplementary Information** file, and the corresponding discussion has been given in the section of Results and discussion (Lines 3-6, Page 15) of revised manuscript. It now reads:

“Furthermore, the therapeutic benefit of our FFs-ICG nanoNETs extends to established infections (3 days) with mature biofilms (Supplementary Fig. 27). This significantly enhances the clinical relevance of our platform, as it demonstrates efficacy not just as a prophylactic or early intervention, but as a potential treatment for ongoing, resistant infections.”

Supplementary Fig. 27

Comment 10: When considering the potential side-effects of the treatment, the authors should add a cytokine profile in plasma, and leukocyte analysis in blood and tissue, in particular for neutrophils and monocytes.

Response: We thank the reviewer for the insightful suggestion. To directly address this point, we have conducted extensive additional experiments to collect the blood samples from the mice in different treatment groups, and perform the hematological analysis (Supplementary Fig. 29). The results demonstrated that key leukocyte parameters, including total white blood cell (WBC) count, lymphocyte count (Lymph), monocyte count (Mon), and neutrophil granulocyte count (Gran), remained within normal physiological ranges in all groups, involved with the FFs-ICG treatment group. Furthermore, other hematological indicators of these treated mice, such as red blood cell (RBC) counts, hemoglobin (HGB) levels, and platelet-related parameters, including hematocrit (HCT), mean corpuscular volume (MCV), platelet count (PLT), mean platelet volume (MPV), platelet distribution width (PDW), and plateletcrit (PCT), also maintained normal level similar with the control group. These data indicate that our FFs-ICG nanoNETs treatment did not elicit significant systemic inflammation or disrupt leukocyte homeostasis, supporting its favorable safety profile.

The new results are summarized in Supplementary Fig. 29 and have been added to the revised manuscript in the section of Results and Discussion (Lines 9-14, Pages 15). It now reads:

“A comprehensive biosafety evaluation, including histological assessment of major organs and blood biochemical analysis, confirmed the excellent *in-vivo* biocompatibility of all investigated biomaterials (Supplementary Fig. 28). Also, the hematological analysis demonstrated that the treatments of these prepared biomaterials, including our FFs-ICG nanoNETs, did not elicit significant systemic inflammation or disrupt leukocyte homeostasis, supporting their favorable safety profiles (Supplementary Fig. 29).”

Supplementary Fig. 29

Comment 11: In supplementary figure 27, the heatmap is not clear. If the data is scaled between 0 and 1, there should be some conditions in red. How is this analyzed? Please, the authors should also show in raw counts. Also, add an explanation of the abbreviations in the legend for readability (check for the rest of the legends as well).

Response: We sincerely appreciate the reviewer's careful attention to these important details. We agree that the previous heatmap was unclear and the normalization approach may have been misleading. In response, we have now re-plotted the blood biochemical data using bar graphs with raw values to more accurately and intuitively represent the results. Additionally, we have carefully reviewed all figure legends in the manuscript and added a full explanation for all abbreviations to improve readability. The revised figure and legends eliminate any potential confusion in data interpretation and ensure clarity for the reader. The revised Supplementary Fig. 28 is shown as follow:

Supplementary Fig. 28

Comment 12: The authors should reduce the number of supplementary figures by combining them.

Response: We thank the reviewer for this helpful suggestion. Consolidating these supplementary figures will improve the clarity and conciseness of the supplementary information. In accordance with this advice, we have thoroughly re-organized the Supplementary Figures by merging related panels into larger, multi-panel figures in the revised **Supplementary Information** file. We trust that since there is not a strict limit on the number on supplementary figures in *Nature Communications*, the final format of the supporting information has sufficient readability.